# LightHGNN: Distilling Hypergraph Neural Networks into MLPs for $100\times$ Faster Inference

**Yifan Feng[1], Yihe Luo[1], Shihui Ying[2], Yue Gao[1]***
[1]School of Software, BNRist, THUIBCS, BLBCI, Tsinghua University
[2]Department of Mathematics, School of Science, Shanghai University
`evanfeng97@gmail.com, luoyh21@mails.tsinghua.edu.cn`
`shying@shu.edu.cn, gaoyue@tsinghua.edu.cn`

## ABSTRACT

Hypergraph Neural Networks (HGNNs) have recently attracted much attention and exhibited satisfactory performance due to their superiority in high-order correlation modeling. However, it is noticed that the high-order modeling capability of hypergraph also brings increased computation complexity, which hinders its practical industrial deployment. In practice, we find that one key barrier to the efficient deployment of HGNNs is the high-order structural dependencies during inference. In this paper, we propose to bridge the gap between the HGNNs and inference-efficient Multi-Layer Perceptron (MLPs) to eliminate the hypergraph dependency of HGNNs and thus reduce computational complexity as well as improve inference speed. Specifically, we introduce LightHGNN and LightHGNN$^+$ for fast inference with low complexity. LightHGNN directly distills the knowledge from teacher HGNNs to student MLPs via soft labels, and LightHGNN$^+$ further explicitly injects reliable high-order correlations into the student MLPs to achieve topology-aware distillation and resistance to over-smoothing. Experiments on eight hypergraph datasets demonstrate that even without hypergraph dependency, the proposed LightHGNNs can still achieve competitive or even better performance than HGNNs and outperform vanilla MLPs by 16.3 on average. Extensive experiments on three graph datasets further show the average best performance of our LightHGNNs compared with all other methods. Experiments on synthetic hypergraphs with 5.5w vertices indicate LightHGNNs can run $100\times$ faster than HGNNs, showcasing their ability for latency-sensitive deployments.

## 1 INTRODUCTION

Compared to the graph with pair-wise correlation, the hypergraph is composed of degree-free hyper-edges, which have an inherent superior modeling ability to represent those more complex high-order correlations. Therefore, many Hypergraph Neural Networks (HGNNs) Feng et al. (2019); Gao et al. (2022); Dong et al. (2020) have been proposed in the vertex classification of citation networks Bai et al. (2021); Yadati et al. (2019), the recommendation in user-item bipartite graphs Xia et al. (2022); Ji et al. (2020), link prediction in drug-protein networks Saifuddin et al. (2023); Ruan et al. (2021), and multi-tissue gene expression imputation Viñas et al. (2023); Murgas et al. (2022). However, for large-scale industrial applications, especially for those big-data, small-memory, and high-speed demand environments, the Multi-Layer Perceptrons (MLPs) remain the primary workhorse. The main reason for such an academic-industrial gap for HGNNs is the dependence on the hypergraph structure in inference, which requires large memories in practice. Consequently, as the scale of the hypergraph and the number of layers of HGNNs increase, the inference time will also exponentially increase, as shown in Figure 1(c), limiting the potential usage of this type of method.

The hypergraph dependence of HGNNs is caused by the high-order neighbor fetching in message passing of vertex-hyperedge-vertex. On the one hand, some GNNs-to-MLPs methods like Graph-Less Neural Networks (GLNN) Zhang et al. (2021) and Knowledge-inspired Reliable Distillation (KRD) Wu et al. (2023) are proposed to distill the knowledge from GNNs to MLPs to eliminate the

---

*Corresponding author: Yue Gao

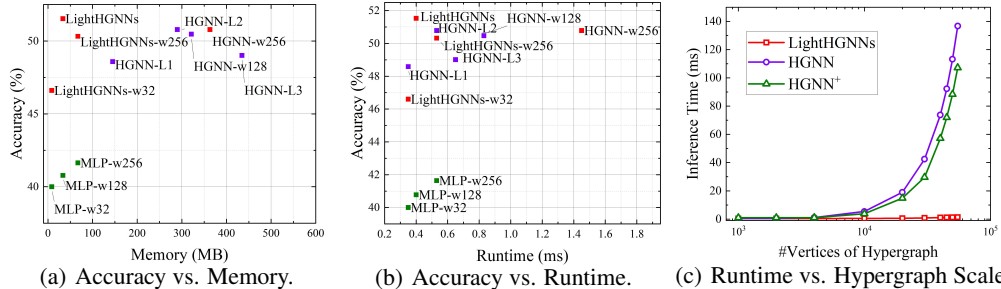

(a) Accuracy vs. Memory.  (b) Accuracy vs. Runtime.  (c) Runtime vs. Hypergraph Scale

Figure 1: Performance and efficiency comparison. (a) and (b) are run on the IMDB-AW hypergraph dataset. (c) provides the inference time comparison on a series of synthetic hypergraph datasets.

graph dependency. Those methods Zhang et al. (2021); Wu et al. (2023) either distill the knowledge by the soft label of teacher GNNs or pull the distance of reliable vertices via pair-wise edge as the extra supervision. Unlike the pair-wise edge in the graph, each hyperedge in the hypergraph can connect more than two vertices. The hypergraph neighborhood is more complex and defined in a hierarchical paradigm Gao et al. (2022). Those GNNs-to-MLPs methods cannot address the high-order correlation in the hypergraph. On the other hand, compared with the HGNNs, the MLPs method performs worse (about averaged 16 decline) on hypergraph datasets, as shown in Tab 1, yet has no dependency on hypergraph structure among batched samples and can be deployed to process any scale of the hypergraph. Upon the above observations, we ask: *Can we fix the gap between MLPs and HGNNs to achieve hypergraph dependency-free inference and topology-aware distillation?*

**Present Work.** In this paper, we proposed the LightHGNN and LightHGNN$^+$ to eliminate the dependence on the hypergraph in the inference of HGNNs and achieve running fast and consuming low memory like MLPs and performing as well as HGNNs as shown in Figure 1(a) and 1(b). The LightHGNN directly distills the knowledge from the teacher HGNNs into the student MLPs with the classical soft label Hinton et al. (2015) of the teacher. The LightHGNN$^+$ further develops a topology-aware knowledge distillation supervised by both the soft labels and the proposed high-order soft labels. To generate the high-order soft labels, we first quantify the reliability of hyperedges by the resistance to noise to generate a reliable hyperedge set. Then, the high-order soft labels can be generated via propagating those soft labels from vertex to hyperedge on those sampled reliable hyperedges. The proposed high-order soft-label constraint can explicitly inject the high-order topology knowledge from those reliable hyperedges into the student MLPs. We further design the topology-aware score to quantify the relevance of features and the topology, which is the main consequence of over-smoothing Cai & Wang (2020). The results indicate the proposed topology-aware distillation can effectively resist the over-smoothing on both graphs and hypergraphs.Experiments on 11 graph and hypergraph datasets demonstrate the effectiveness of our LightHGNNs compared to GNNs, HGNNs, and GNNs-to-MLPs. Experiments on eight hypergraph datasets indicate that the proposed LightHGNNs can significantly outperform the MLPs with averaged 16.3 improvements and only have a slight performance decline of 0.39 compared to the HGNN. Experiments on the synthetic hypergraph datasets indicate that LightHGNNs run 100× faster than the HGNN and can potentially deploy in the large-scale hypergraph dataset. Our main contributions are as follows:

- We propose the LightHGNNs to bridge the gap between MLPs and HGNNs to achieve hypergraph dependency-free inference and topology-aware distillation.
- We design the reliable hyperedge quantification and sampling strategy to achieve topology-aware distillation, which can explicitly inject reliable high-order knowledge into student MLPs and achieve an averaged 16.3 performance gain compared to the vanilla MLPs.
- We develop the topology-aware score to quantify the over-smoothing phenomenon from the relevance of features and topology perspective and demonstrate the proposed LightHGNN$^+$ can effectively resist over-smoothing.
- We show that the proposed LightHGNNs can run 100x faster than the HGNN, which can be easily deployed in latency-constrained applications.

## 2 RELATED WORK

**Hypergraph Neural Networks.** The early HGNNs are defined upon the spectral domain like HGNN Feng et al. (2019) and HpLapGCN Fu et al. (2019), which conduct the feature smoothing via the hypergraph Laplacian matrix. Yadati et al. (2019) propose the HyperGCN, which designs a

strategy to reduce the hypergraph to graphs and further employ the GNNs Kipf & Welling (2017) to learn the representations. Besides, a series of spatial-based hypergraph convolutions are proposed, like the vertex-hyperedge attention mechanism (Bai et al., 2021), dynamic hypergraph construction (Jiang et al., 2019), and two-stage message passing (Gao et al., 2022; Dong et al., 2020).

**GNNs-to-MLPs Distillation.** To enjoy the efficiency and effectiveness of MLPs and GNNs, GLNN (Zhang et al., 2021) directly utilizes the prediction distribution of teacher GNNs as the soft target to supervise the student MLPs, which ignores the topology of the original graph in distillation. Yang et al. (2021) extracts the knowledge of an arbitrary learned GNN model (teacher model) and injects it into a well-designed student model to achieve more efficient predictions. KRD (Wu et al., 2023) further quantifies the knowledge of each vertex and pulls the distance between it and its neighbors.Existing methods are constrained in the low-order graph neural networks. In contrast, this paper aims to bridge the hypergraph neural networks and MLPs and design topology-aware distillation, injecting reliable high-order knowledge into MLPs for faster inference than HGNNs.

## 3 PRELIMINARIES

**Notions and Problem Statement.** Let $\mathcal{G} = \{\mathcal{V}, \mathcal{E}\}$ be a hypergraph, set $\mathcal{V}$ and $\mathcal{E}$ denote the vertex set and hyperedge set, respectively. $N = |\mathcal{V}|$ and $M = |\mathcal{E}|$. Each vertex $\boldsymbol{v}_i$ is associated with a feature vector $\boldsymbol{x}_i$, and the overall vertex feature matrix is denoted by $\boldsymbol{X} = [\boldsymbol{x}_1, \boldsymbol{x}_2, \cdots, \boldsymbol{x}_N] \in \mathbb{R}^{N \times c}$. In practice, the hypergraph can be represented by an incidence matrix $\boldsymbol{H} \in \{0, 1\}^{N \times M}$, where the row and column denote the vertices and hyperedges, respectively. $\boldsymbol{H}(v, e) = 1$ denotes the vertex $v$ belongs to the hyperedge $e$. Besides, $\mathcal{Y} = \{\boldsymbol{y}_v \mid v \in \mathcal{V}\}$ denotes the vertex label set. Consider a semi-supervised vertex classification task, the vertex set $\mathcal{V}$ is divided into two sub-sets: labeled data $\mathcal{D}^L = (\mathcal{V}^L, \mathcal{Y}^L)$ and the unlabeled data $\mathcal{D}^U = (\mathcal{V}^U, \mathcal{Y}^U)$. The task aims to learn a map $\phi : \mathcal{V} \to \mathcal{Y}$, which can be used to predict the label of those unlabeled vertices.

**Hypergraph Neural Networks (HGNNs).** HGNN is defined based on the hypergraph Laplacian matrix, and its eigenvectors are treated as the Fourier bases for a given signal $\boldsymbol{x}$. Furthermore, the Chebyshv polynomial Defferrard et al. (2016) is adopted for the convenience of computation. Then, the HGNN can be defined as: $\boldsymbol{X}^{l+1} = \sigma(\boldsymbol{D}_v^{-1/2} \boldsymbol{H} \boldsymbol{W} \boldsymbol{D}_e^{-1} \boldsymbol{H}^\top \boldsymbol{D}_v^{-1/2} \boldsymbol{X}^t \boldsymbol{\Theta})$, where $\boldsymbol{D}_v$ and $\boldsymbol{D}_e$ are the diagonal matrix of the degree of vertices and hyperedges, respectively. $\boldsymbol{W}$ is the diagonal matrix of the weight of hyperedges, and $\boldsymbol{\Theta}$ is the trainable parameters.

## 4 METHODOLOGY

In this section, we introduce LightHGNN and LightHGNN$^+$. Then, we provide the complexity analysis and discuss the relationship to existing GNNs-to-MLPs methods.

### 4.1 LIGHTHGNN: SOFT-TARGET GUIDED HGNNs DISTILLATION

To boost the inference performance in real-world deployment, we propose the soft-target guided HGNNs distillation, named LightHGNN, which directly distills the knowledge of HGNNs into the MLPs. Motivated by the Knowledge Distillation (KD) Hinton et al. (2015) and GNNs-to-MLPs methods Zhang et al. (2021); Wu et al. (2023), we adopt the MLPs as the student network and the well-trained HGNNs as the teacher network, and distills the knowledge with the combination objective $\mathcal{L}_{DH}$ of the Cross-Entropy loss $\mathcal{L}_{ce}$ and the Kullback-Leibler Divergence loss as follows:

$$\mathcal{L}_{DH} = \lambda \frac{1}{|\mathcal{V}^L|} \sum_{v \in \mathcal{V}^L} \mathcal{L}_{ce}(\hat{\boldsymbol{y}}_v^s, \boldsymbol{y}_v) + (1 - \lambda) \frac{1}{|\mathcal{V}|} \sum_{v \in \mathcal{V}} D_{\text{KL}}(\hat{\boldsymbol{y}}_v^s, \hat{\boldsymbol{y}}_v^t), \tag{1}$$

where $\boldsymbol{y}_v$ is the one-hot encoded label of vertex $v$. Vector $\hat{\boldsymbol{y}}_v^t$ and $\hat{\boldsymbol{y}}_v^s$ are the softmax normalized prediction of vertex $v$ from the teacher HGNNs and student MLPs, respectively. Note that the first term of the $\mathcal{L}_{DH}$ only computes the typical cross-entropy loss on labeled set $\mathcal{V}^L$. The second term pulls the distance between the soft target of teacher HGNNs and the prediction distribution of student MLPs on the vertex set $\mathcal{V}$. The hyper-parameter $\lambda$ is adopted to balance the two terms. The model essentially is the MLPs with cross-entropy and soft target supervision. Thus, LightHGNN has no dependency on the hypergraph structure and, during inference, runs as fast as MLPs.

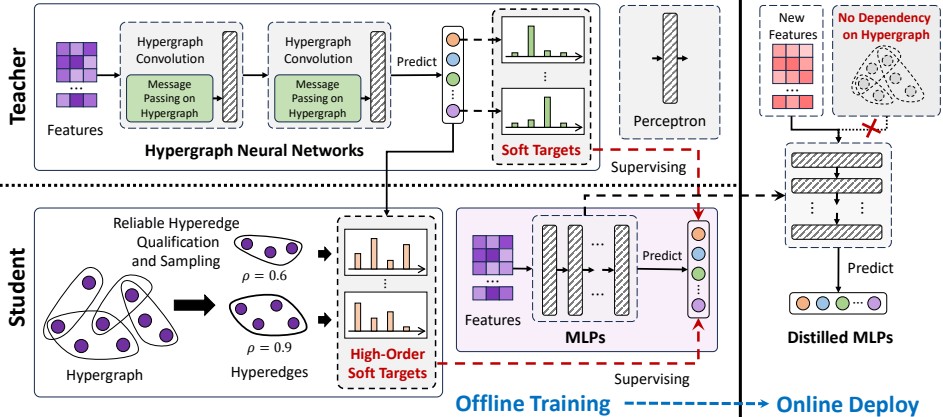

Figure 2: The framework of the proposed Distilled Hypergraph Neural Networks (LightHGNN$^+$).

## 4.2 LIGHTHGNN$^+$: RELIABLE HYPEREDGE GUIDED TOPOLOGY-AWARE DISTILLATION

The LightHGNN simply injects the soft-target information into the student MLPs and ignores the high-order correlations from the original hypergraph. Therefore, we further propose the topology-aware distillation, named LightHGNN$^+$, as illustrated in Figure 2. In the figure, the left part is the offline training stage, and the right part is the online deployment with the distilled MLPs. The top of the left part is the teacher HGNNs, which takes the vertex features and the hypergraph structure as inputs and is supervised by the true label of these labeled vertexes. The bottom of the left part is the student MLPs, which only use the vertex features as input. It is supervised by the true label of the labeled vertices, the soft target of the output of the teacher HGNNs, and the high-order soft target of those reliable hyperedges. In the following, we will introduce how to qualify the reliability of hyperedges, the probability model of sampling reliable hyperedges, the extra high-order soft target constraint for the distillation, and the loss function of the proposed LightHGNN$^+$.

### 4.2.1 RELIABLE HYPEREDGE QUALIFICATION

Hyperedges, as the core of a hypergraph, can represent the high-order correlation among vertices. However, not all hyperedges can provide reliable information for the downstream task. Thus, we develop an entropy-based reliable hyperedge qualification method to **quantify the relevance of the hyperedge to the task.** as depicted in Figure 3. Given the well-trained teacher HGNNs $f_\theta$ : $(\boldsymbol{X}, \boldsymbol{H}) \to \boldsymbol{Y}$, we add noise $\epsilon$ on the input features and measure the invariance of the hyperedge entropy to determine the reliability of hyperedges, as follows:

$$\delta_e = \mathop{\mathbb{E}}_{\epsilon \sim \mathcal{N}(\boldsymbol{\mu}, \boldsymbol{\Sigma})} \left\| \frac{1}{|e|} \sum_{v \in e} \mathcal{H}(\hat{\boldsymbol{y}}'_v) - \frac{1}{|e|} \sum_{v \in e} \mathcal{H}(\hat{\boldsymbol{y}}_v) \right\|_2, \tag{2}$$
$$\text{where} \quad \boldsymbol{Y}' = f_\theta(\epsilon \boldsymbol{X}, \boldsymbol{H}) \quad \text{and} \quad \boldsymbol{Y} = f_\theta(\boldsymbol{X}, \boldsymbol{H})$$

where $\mathcal{H}(\boldsymbol{p}) = -\sum_i p_i \log(p_i)$ is the information entropy. Given a hyperedge, we calculate the average entropy of its connected vertices' prediction distribution. The variance $\delta_e$ of the average entropy of the hyperedge after introducing the noise $\epsilon \sim \mathcal{N}(\boldsymbol{\mu}, \boldsymbol{\Sigma})$ is used to compute the hyperedge's reliable score $\rho_e$. **The larger value of $\delta_e$ indicates the hyperedge is more sensitive to the noise perturbation in the downstream task**. Then, we normalize the $\delta_e$ with the max value and compute the reliable score of hyperedge with $\rho_e = 1 - \frac{\delta_e}{\delta_{\max}}$. Clearly, the $\rho_e$ measures the robustness of hyperedge $e$ connected vertices of teacher HGNNs to noise perturbation and reflects the reliability of hyperedge with respect to the downstream task. Those hyperedges with higher reliable scores containing robust knowledge should be paid more attention in the distillation process.

### 4.2.2 SAMPLING PROBABILITY MODELING AND HIGH-ORDER SOFT-TARGET CONSTRAINT

To fully use those hyperedge reliable scores, we propose a sampling probability modeling for hyperedge selection and develop high-order soft target constraint as an additional supervision for high-order topology-awareness distillation as shown in Figure S1. Here, the Bernoulli distribution is adopted to model the hyperedge sampling as: $p(s_i \mid \rho_{e_i}) \sim \text{Bernoulli}(\rho_{e_i})$ and $e_i \in \mathcal{E}$, where $s_i$ is the sampling probability of the hyperedge $e_i \in \mathcal{E}$. Given the hypergraph $\mathcal{G} = \{\mathcal{V}, \mathcal{E}\}$, a sub-set,

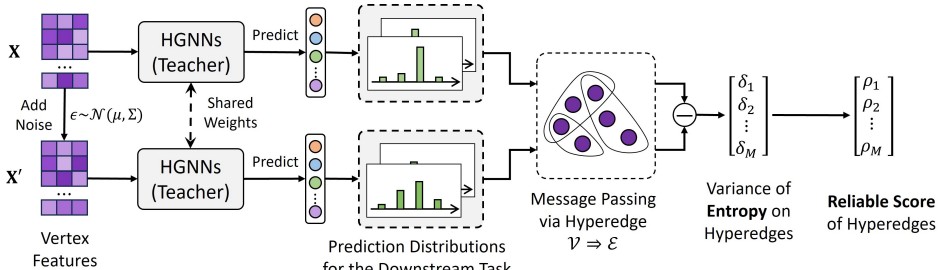

Figure 3: The illustration of reliable hyperedge qualification.

named reliable hyperedge set $\mathcal{E}'$, is drawn with independent Bernoulli distribution by the specific parameter $\rho_e$ for each hyperedge, as shown in Figure S1 (a) (Appendix). Each hyperedge may contain vertices from different categories in the hypergraph and have the unique property behind the high-order correlation. Directly distilling the knowledge from the vertex's soft target may lose the crucial high-order correlation information. Thus, we propose constructing the high-order soft target via the vertex's soft target and those reliable hyperedges to **inject the reliable high-order information into the distilled MLPs**, as shown in Figure S1 (b) (Appendix). Given the soft target set $\{\boldsymbol{y}_v^t \mid v \in \mathcal{V}\}$ and reliable hyperedge set $\mathcal{E}'$, the high-order soft target can be computed via a naive message passing from vertex to hyperedge as follows:

$$\boldsymbol{z}_e^s = \frac{1}{|e|} \sum_{v \in \mathcal{N}_v(e)} \hat{\boldsymbol{y}}_v^s \quad \text{and} \quad \boldsymbol{y}_e^t = \frac{1}{|e|} \sum_{v \in \mathcal{N}_v(e)} \hat{\boldsymbol{y}}_v^t \quad \text{for} \quad e \in \mathcal{E}', \tag{3}$$

where $\mathcal{N}_v(e)$ denotes a set of the connected vertices of the hyperedge $e$. $\boldsymbol{y}_e^t$ and $\boldsymbol{z}_e^s$ denote the high-order soft target and the predicted high-order distribution, respectively. Then, the additional high-order soft-target constraint can be achieved by the KD Divergence as follows:

$$\mathcal{L}_{hc} = \frac{1}{|\mathcal{E}'|} \sum_{\substack{e \in \mathcal{E}' \\ e_i \sim p(s_i | \rho_{e_i})}} D_{\mathrm{KL}}(\alpha(\boldsymbol{z}_e^s / \tau), \alpha(\boldsymbol{y}_e^t / \tau)), \tag{4}$$

where $\alpha(\cdot)$ is the Softmax function for normalization, and $\tau$ is the distillation temperature coefficient. The $\mathcal{L}_{hc}$ is designed to pull the distance between the predicted high-order distribution of the student MLPs and the high-order soft label of the teacher HGNNs. By minimizing the $\mathcal{L}_{hc}$, the distilled MLPs can reserve more reliable high-order information to achieve better performance. Finally, the total loss function of LightHGNN$^+$ is formulated as follows:

$$\mathcal{L}_{DH^+} = \lambda \frac{1}{|\mathcal{V}^L|} \sum_{v \in \mathcal{V}^L} \mathcal{L}_{ce}(\hat{\boldsymbol{y}}_v^s, \boldsymbol{y}_v) + (1 - \lambda) \left( \frac{1}{|\mathcal{V}|} \sum_{v \in \mathcal{V}} D_{\mathrm{KL}}(\hat{\boldsymbol{y}}_v^s, \hat{\boldsymbol{y}}_v^t) + \mathcal{L}_{hc} \right), \tag{5}$$

where $\lambda$ is a hyper-parameter to balance the information from the true labels ($\boldsymbol{y}_v$) and teacher HGNNs (the soft targets $\hat{\boldsymbol{y}}_v^t$ and high-order soft targets $\boldsymbol{y}_e^t$). Note that the supervision information of the true labels is on the labeled vertex set $\mathcal{V}^L$. The supervision information of soft targets and high-order soft targets are on the entire vertex set $\mathcal{V}$ and reliable hyperedge set $\mathcal{E}'$, respectively.

## 4.3 ANALYSIS

**Time Complexity Analysis.** The pseudo-code of LightHGNN$^+$ framework is in Appendix A. The training of LightHGNNs can be divided into two steps: pre-training the teacher HGNNs and distilling knowledge into student MLPs, which are supervised by the cross-entropy loss and distillation loss, respectively. The training and inference complexity comparison is provided in Appendix B.

**Relation to GNNs-to-MLPs Methods.** We further discuss the relationship between the proposed method and the related GNNs-to-MLPs methods Zhang et al. (2021); Wu et al. (2023). The Graph-Less Neural Networks (GLNN) Zhang et al. (2021) proposes using the soft label of the teacher GNNs to supervise the student MLPs. Compared to it, our LightHGNN is a simple extension from GNNs to HGNNs. However, our LightHGNN$^+$ further proposes the high-order soft label to help the student MLPs learn more high-order structure information from the teacher HGNNs. As for the Knowledge-inspired Reliable Distillation (KRD) Wu et al. (2023), its student MLPs can only be supervised by the soft label from those reliable vertices (knowledge point in their paper), which still

lose the structure information and cannot be utilized in the hypergraph. However, our LightHGNN$^+$ can quantify the reliability of the high-order correlations and inject the reliable high-order information into the student MLPs via explicit supervision from high-order soft labels of those reliable hyperedges. The proposed LightHGNN$^+$ can sufficiently take advantage of both the vertex features and high-order structure information, thus yielding better performance and faster inference speed.

## 5 EXPERIMENTS

**Datasets, Compared Methods, and Settings.** In our experiments, we adopt three typical graph datasets: Cora Sen et al. (2008), Pubmed McCallum et al. (2000), and Citeseer Giles et al. (1998), and eight hypergraph datasets: News20 Asuncion & Newman (2007), CA-Cora, CC-Cora, CC-Citeseer Yadati et al. (2019), DBLP-Paper, DBLP-Term, DBLP-Conf Sun et al. (2011), and IMDB-AW Fu et al. (2020). More details of the datasets are in Appendix C. We compare three types of methods: GNNs (including GCN Kipf & Welling (2017) and GAT Velickovic et al. (2017)), HGNNs (including HGNN Feng et al. (2019), HGNN$^+$ Gao et al. (2022), and HNHN Dong et al. (2020)), GNNs-to-MLPs (including GLNN Zhang et al. (2021), KRD Wu et al. (2023) and NOSMOG Tian et al. (2022)). We adopt two experimental settings for sufficient comparison, including the transductive and production settings. The transductive is a classical setting for vertex classification on graph/hypergraph datasets. The production setting contains both the transductive and inductive predictions, which is developed to evaluate the performance towards realistic deployment. More details about the two settings and splitting of different datasets are provided in Appendix D. We run 5 times with different random seeds for each experiment and report the average and standard deviation. The hyper-parameter configuration of each method is provided in Appendix E.

Table 1: Experimental results on eight hypergraph datasets under transductive setting.

| Dataset | MLP | HGNN | LightHGNN | LightHGNN$^+$ | $\Delta_{MLP}$ | $\Delta_{HGNN}$ |
|---|---|---|---|---|---|---|
| *News20* | $63.53_{\pm2.70}$ | $\underline{75.94}_{\pm0.66}$ | $\mathbf{76.25}_{\pm0.32}$ | $75.87_{\pm0.38}$ | 12.72 | +1.70 |
| *CA-Cora* | $51.86_{\pm1.47}$ | $73.19_{\pm1.12}$ | $\mathbf{73.63}_{\pm5.14}$ | $\underline{73.54}_{\pm3.80}$ | 21.76 | +0.44 |
| *CC-Cora* | $49.87_{\pm1.38}$ | $68.21_{\pm3.68}$ | $\underline{69.25}_{\pm3.46}$ | $\mathbf{69.48}_{\pm3.04}$ | 19.61 | +1.27 |
| *CC-Citeseer* | $51.79_{\pm2.59}$ | $\mathbf{63.37}_{\pm2.63}$ | $62.97_{\pm3.53}$ | $\underline{63.01}_{\pm2.76}$ | 11.27 | −0.30 |
| *DBLP-Paper* | $62.84_{\pm1.58}$ | $72.27_{\pm0.91}$ | $\underline{72.74}_{\pm1.07}$ | $\mathbf{72.93}_{\pm0.97}$ | 10.09 | +0.66 |
| *DBLP-Term* | $62.84_{\pm1.58}$ | $\mathbf{82.01}_{\pm2.27}$ | $80.27_{\pm0.91}$ | $\underline{80.77}_{\pm0.68}$ | 17.93 | −1.23 |
| *DBLP-Conf* | $62.84_{\pm1.58}$ | $\mathbf{94.07}_{\pm0.13}$ | $90.23_{\pm0.58}$ | $\underline{90.24}_{\pm0.71}$ | 27.40 | −3.83 |
| *IMDB-AW* | $40.87_{\pm1.43}$ | $\underline{50.47}_{\pm1.66}$ | $50.19_{\pm1.56}$ | $\mathbf{50.78}_{\pm1.92}$ | 9.90 | +0.30 |
| *Avg. Rank/Avg.* | 4.0 | 1.7 | 2.4 | 1.8 | 16.33 | −0.29 |

### 5.1 HOW DO LIGHTHGNNS COMPARE TO MLPS AND HGNNS?

We start by comparing the LightHGNNs to MLPs and HGNNs on eight hypergraph datasets under standard transductive learning. As shown in Table 1, the performance of LightHGNNs improves over MLPs by large margins of about averaged 16.3. Compared with the HGNN, the proposed LightHGNNs exhibit a slight performance degradation of about 0.29. Besides, we observe that the proposed LightHGNN$^+$ obtains seven times the best or second-best performance and ranks very close to the teacher HGNN on eight hypergraph datasets. However, the LightHGNNs adopt the same architecture as MLPs without hypergraph dependency. The experimental results demonstrate the effectiveness of distilling the knowledge from teacher HGNNs to student MLPs. We attribute the improvement of LightHGNN$^+$ to the devised topology-aware distillation, which can further extract those reliable hyperedges and explicitly inject the reliable topology knowledge into the student MLPs, thus yielding better performance than the LightHGNN without topology distillation.

In real-world applications, not all samples can be seen in the training phase. Thus, to fully validate the performance of the proposed methods confronting realistic deployment, we adopt a more general setting: the production setting, which contains both transductive and inductive prediction. More details can be found in Appendix D. As shown in Table 2, we see that the LightHGNNs still outperform MLPs by large margins of about 14.8. However, we also notice the distinct margin of the HGNN and LightHGNNs under the inductive setting, especially on the DBLP-Conf dataset, about 11% decline. This is because the dataset only contains 20 hyperedges with averaged linking 982.2 vertices, which leads to a high dependency on the topology of the prediction. In the inductive setting, HGNN

Table 2: Experimental results on eight hypergraph datasets under production setting.

| Dataset | Setting | MLPs | HGNN | LightHGNN | LightHGNN$^+$ | $\Delta_{MLP}$ | $\Delta_{HGNN}$ |
|---|---|---|---|---|---|---|---|
| *News20* | Prod. | $63.86_{\pm3.01}$ | $\underline{75.18}_{\pm1.65}$ | $\mathbf{75.81}_{\pm1.28}$ | $74.56_{\pm1.34}$ | 11.95 | +0.63 |
| | Tran. | $63.80_{\pm3.03}$ | $\underline{75.08}_{\pm1.44}$ | $\mathbf{75.81}_{\pm1.23}$ | $74.58_{\pm1.24}$ | 12.01 | +0.73 |
| | Ind. | $64.10_{\pm2.97}$ | $75.13_{\pm2.01}$ | $\mathbf{75.82}_{\pm1.61}$ | $\underline{75.44}_{\pm1.94}$ | 11.71 | +0.68 |
| *CA-Cora* | Prod. | $50.73_{\pm1.43}$ | $\underline{71.01}_{\pm3.19}$ | $70.33_{\pm3.49}$ | $\mathbf{71.46}_{\pm2.13}$ | 20.72 | +0.45 |
| | Tran. | $50.75_{\pm1.64}$ | $70.80_{\pm3.25}$ | $\underline{71.62}_{\pm4.29}$ | $\mathbf{72.49}_{\pm2.13}$ | 21.73 | +1.68 |
| | Ind. | $50.67_{\pm1.44}$ | $\mathbf{70.83}_{\pm2.83}$ | $65.14_{\pm2.95}$ | $\underline{67.34}_{\pm3.36}$ | 16.67 | −3.48 |
| *CC-Cora* | Prod. | $50.73_{\pm1.43}$ | $\underline{68.20}_{\pm3.89}$ | $\mathbf{68.29}_{\pm4.47}$ | $67.89_{\pm3.58}$ | 17.55 | +0.09 |
| | Tran. | $50.75_{\pm1.64}$ | $68.26_{\pm3.92}$ | $\mathbf{69.00}_{\pm4.16}$ | $\underline{68.70}_{\pm3.32}$ | 18.24 | +0.73 |
| | Ind. | $50.67_{\pm1.44}$ | $\mathbf{66.00}_{\pm4.55}$ | $\underline{65.46}_{\pm5.87}$ | $64.61_{\pm4.69}$ | 14.79 | −0.53 |
| *CC-Citeseer* | Prod. | $54.41_{\pm1.36}$ | $\underline{64.02}_{\pm0.92}$ | $62.90_{\pm1.95}$ | $\mathbf{64.11}_{\pm0.85}$ | 9.69 | +0.09 |
| | Tran. | $54.42_{\pm1.52}$ | $\underline{63.74}_{\pm0.75}$ | $63.30_{\pm1.92}$ | $\mathbf{64.53}_{\pm0.63}$ | 10.10 | +0.79 |
| | Ind. | $54.36_{\pm1.14}$ | $\mathbf{63.51}_{\pm1.34}$ | $61.31_{\pm2.21}$ | $\underline{61.93}_{\pm2.01}$ | 7.56 | −1.58 |
| *DBLP-Paper* | Prod. | $63.23_{\pm1.48}$ | $\underline{71.52}_{\pm1.31}$ | $71.14_{\pm1.23}$ | $\mathbf{71.69}_{\pm1.44}$ | 8.46 | +0.16 |
| | Tran. | $62.97_{\pm1.69}$ | $70.75_{\pm1.49}$ | $\underline{70.88}_{\pm1.29}$ | $\mathbf{71.40}_{\pm1.50}$ | 8.42 | +0.65 |
| | Ind. | $64.25_{\pm1.75}$ | $\underline{72.72}_{\pm2.32}$ | $72.22_{\pm2.08}$ | $\mathbf{72.86}_{\pm2.33}$ | 8.61 | +0.14 |
| *DBLP-Term* | Prod. | $63.56_{\pm1.15}$ | $\mathbf{81.08}_{\pm2.51}$ | $\underline{78.39}_{\pm3.22}$ | $78.32_{\pm2.70}$ | 14.83 | −2.69 |
| | Tran. | $63.37_{\pm1.17}$ | $\mathbf{81.23}_{\pm2.39}$ | $78.54_{\pm3.08}$ | $\underline{78.58}_{\pm2.73}$ | 15.21 | −2.64 |
| | Ind. | $64.30_{\pm1.50}$ | $\mathbf{81.56}_{\pm2.75}$ | $\underline{77.79}_{\pm4.15}$ | $77.28_{\pm3.29}$ | 13.48 | −3.77 |
| *DBLP-Conf* | Prod. | $63.56_{\pm1.15}$ | $\mathbf{94.15}_{\pm0.19}$ | $89.48_{\pm0.52}$ | $\underline{89.50}_{\pm0.49}$ | 25.94 | −4.64 |
| | Tran. | $63.37_{\pm1.17}$ | $\mathbf{94.08}_{\pm0.32}$ | $91.12_{\pm0.76}$ | $\underline{91.20}_{\pm0.74}$ | 27.83 | −2.87 |
| | Ind. | $64.30_{\pm1.50}$ | $\mathbf{94.21}_{\pm0.52}$ | $\underline{82.93}_{\pm1.15}$ | $82.68_{\pm0.57}$ | 18.62 | −11.27 |
| *IMDB-AW* | Prod. | $41.05_{\pm2.49}$ | $\mathbf{50.29}_{\pm1.58}$ | $49.10_{\pm1.64}$ | $\underline{49.12}_{\pm2.02}$ | 8.05 | −1.17 |
| | Tran. | $41.16_{\pm2.67}$ | $\underline{49.46}_{\pm1.43}$ | $49.39_{\pm1.63}$ | $\mathbf{49.68}_{\pm1.86}$ | 8.51 | +0.21 |
| | Ind. | $40.61_{\pm1.95}$ | $\mathbf{52.05}_{\pm2.68}$ | $\underline{47.96}_{\pm2.12}$ | $47.58_{\pm3.21}$ | 7.35 | −4.08 |
| *Avg. Rank/Avg.* | | 4.0 | 1.5 | 2.3 | 1.9 | 14.86 | −1.45 |

can utilize the extra topology information of those unseen vertices (unseen topology) to support the prediction, while LightHGNN cannot. Therefore, the LightHGNN shows a distinct performance decline. It is essential to distill the general topology knowledge and learn the topology-aware ability towards the unseen topology under the inductive setting, which can be exploited in further work.

## 5.2 HOW DO LIGHTHGNNS COMPARE TO GNNS AND GNNS-TO-MLPS?

We further compare LightHGNNs to existing GNNs and GNNs-to-MLPs on both graph and hypergraph datasets. More details of the experimental setting are in Appendix F. As shown in Table 3, unlike the results on pure hypergraph datasets (Tables 1 and 2), the LightHGNN$^+$ achieves the average first place, which outperforms MLPs, GNNs, HGNNs, and GNNs-to-MLPs methods. The LightHGNNs show comparable performance to HGNNs in the hypergraph datasets while showing better performance in the graph datasets. This is because the topology-aware distillation can adaptively select the task-relevant low-order and high-order structures as extra supervision. As demonstrated in HGNN$^+$ Gao et al. (2022), explicitly modeling those potential high-order structures can improve the model's performance, especially those graph datasets lacking high-order correlations.

Table 3: Experimental results on graph and hypergraph datasets.

| Type | Model | Graph Datasets | | | Hypergraph Datasets | | | Avg. Rank |
|---|---|---|---|---|---|---|---|---|
| | | *Cora* | *Pubmed* | *Citeseer* | *CA-Cora* | *DBLP-Paper* | *IMDB-AW* | |
| MLPs | MLP | $49.64_{\pm1.13}$ | $66.05_{\pm2.78}$ | $51.69_{\pm2.08}$ | $51.86_{\pm1.47}$ | $62.84_{\pm1.58}$ | $40.87_{\pm1.43}$ | 10.0 |
| GNNs | GCN | $79.90_{\pm1.75}$ | $77.54_{\pm1.63}$ | $69.58_{\pm1.89}$ | $72.82_{\pm1.70}$ | $72.02_{\pm1.43}$ | $50.62_{\pm1.44}$ | 5.3 |
| | GAT | $78.35_{\pm2.24}$ | $76.54_{\pm1.56}$ | $69.38_{\pm2.33}$ | $70.73_{\pm1.75}$ | $72.53_{\pm1.15}$ | $49.55_{\pm1.82}$ | 7.7 |
| HGNNs | HGNN | $80.04_{\pm1.42}$ | $76.93_{\pm1.38}$ | $69.89_{\pm1.94}$ | $73.19_{\pm1.12}$ | $72.27_{\pm0.91}$ | $50.47_{\pm1.66}$ | 5.3 |
| | HGNN$^+$ | $78.75_{\pm1.44}$ | $77.54_{\pm1.63}$ | $69.15_{\pm2.08}$ | $72.79_{\pm1.28}$ | $\mathbf{73.05}_{\pm1.69}$ | $\underline{50.67}_{\pm1.75}$ | 5.2 |
| GNNs-to-MLPs | GLNN | $\mathbf{80.93}_{\pm1.90}$ | $78.36_{\pm1.99}$ | $69.88_{\pm1.66}$ | $72.19_{\pm3.83}$ | $72.50_{\pm1.62}$ | $50.48_{\pm1.51}$ | 4.3 |
| | KRD | $79.47_{\pm1.73}$ | $78.72_{\pm1.94}$ | $69.82_{\pm3.36}$ | $71.75_{\pm3.53}$ | $72.85_{\pm6.76}$ | $49.65_{\pm2.12}$ | 5.5 |
| | NOSMOG | $80.12_{\pm0.91}$ | $\mathbf{80.42}_{\pm0.33}$ | $\mathbf{70.86}_{\pm3.53}$ | $68.96_{\pm7.34}$ | $71.47_{\pm2.13}$ | $48.96_{\pm1.43}$ | 5.5 |
| HGNNs-to-MLPs | LightHGNN | $80.36_{\pm2.06}$ | $79.15_{\pm1.57}$ | $69.17_{\pm3.27}$ | $\mathbf{73.63}_{\pm5.14}$ | $72.74_{\pm1.07}$ | $50.19_{\pm1.56}$ | 4.2 |
| | LightHGNN$^+$ | $\underline{80.68}_{\pm1.74}$ | $\underline{79.16}_{\pm1.37}$ | $\underline{70.34}_{\pm1.95}$ | $\underline{73.54}_{\pm3.80}$ | $\underline{72.93}_{\pm0.97}$ | $\mathbf{50.78}_{\pm1.92}$ | 1.8 |

### 5.3 How Fast are LightHGNNs compared to MLPs and HGNNs?

In the deployment environment, the timeliness of the model is crucial. A good model can achieve higher performance in less time. Thus, we conduct three experiments for the performance and efficiency comparison as shown in Figure 1. Figures 1(a) and 1(b) show the comparison of accuracy vs. memory and runtime on the IMDB-AW dataset, respectively. The suffixes "-w32" and "-L3" represent the dimension of the hidden feature and the number of layers of the model, respectively. The upper left is the ideal model with lowe memory, less time, and high accuracy. Obviously, the MLPs runs faster and cost lower memory but has lower accuracy. In contrast, the HGNN performs better but runs slower and consumes more memory. The proposed methods bridge the gap between MLPs and HGNN and have advantages in memory, runtime, and accuracy, as shown in the upper left of the two figures. Considering that the IMDB-AW only contains $4278$ vertices and $5257$ hyperedges, we further generate a series of larger hypergraphs to investigate how fast the proposed LightHGNNs compared to the HGNNs, as shown in Figure 1(c). The x-coordinate with the log scale represents the number of vertices in the hypergraph, and the y-coordinate indicates the inference time of different models. Obviously, under logarithmic scaling of the hypergraph, the runtime of HGNN and $HGNN^+$ increases exponentially in time complexity, while the proposed LightHGNNs still run very fast ($100\times$ faster in the hypergraph with 5.5w vertices compared to the HGNN) and exhibit robust linear time complexity. More details of the three experiments can be found in Appendix G.

### 5.4 How LightHGNN$^+$ Benefits from Topology-Aware Distillation?

As shown in Tables 1, 2 and 3, GNNs and HGNNs significantly outperform the MLPs on the vertex classification task, and LightHGNN exhibits comparable performance to them. With extra topology-aware distillation, LightHGNN$^+$ is often better than LightHGNN. We further investigate how LightHGNN$^+$ benefits from topology-aware distillation. As we all know, GNNs and HGNNs often suffer from over-smoothing Cai & Wang (2020); Chen et al. (2022), which means the **higher relevance of the vertex features and the topology**. This is because that task-irrelevant information Wu et al. (2020); Zhang et al. (2022), including connections and features, will amplify the noise on vertices as layers go deeper. In our topology-aware distillation, only a few task-relevant hyperedges are selected to inject reliable topology knowledge into the student MLPs. Here, we design a topology-aware score $\mathcal{S}$ to measure the relevance of features and hypergraph typology as:

$$\mathcal{S} = \frac{1}{|\mathcal{E}|} \sum_{e \in \mathcal{E}} \frac{\sum_{v \in e} ||\boldsymbol{x}_v - \boldsymbol{x}_e||_2}{d_e(e)} \quad \text{and} \quad \boldsymbol{x}_e = \frac{1}{d_e(e)} \sum_{v \in e} \boldsymbol{x}_v, \tag{6}$$

where $\boldsymbol{x}_v$ and $\boldsymbol{x}_e$ are the embeddings of vertex $v$ and hyperedge $e$, respectively. The $d_e(e)$ denotes the degree of the hyperedge $e$. The vertex embedding is the output of the first convolution layer, and the hyperedge embedding is calculated by aggregating the embedding of its connected vertices. Then, the topology-aware score measures the average distance of vertex and hyperedge. The lower score indicates the vertex feature is closer in each hyperedge and more relevant to the topology. The higher score indicates a lower relevance to the topology.

Table 4: The topology-aware score comparison in graph and hypergraph datasets.

| Model | Graph Datasets | | | Hypergraph Datasets | | | Avg. |
|---|---|---|---|---|---|---|---|
| | Cora | Pubmed | Citeseer | CA-Cora | DBLP-Paper | IMDB-AW | |
| MLPs | 3.78 | 2.06 | 2.64 | 2.76 | 1.02 | 1.18 | 2.24 |
| GCN | 0.25 | 0.32 | 0.09 | 0.08 | 0.12 | 0.08 | 0.15 |
| HGNN | 0.31 | 0.27 | 0.13 | 0.10 | 0.12 | 0.04 | 0.16 |
| LightHGNN | 1.20 | 1.50 | 0.33 | 0.94 | 0.69 | 0.71 | 0.89 |
| LightHGNN$^+$ | 1.58 | 1.95 | 0.64 | 1.15 | 0.73 | 0.77 | 1.14 |

The topology-aware scores of different models on graph and hypergraph datasets are shown in Table 4. Since the vanilla MLPs do not utilize the topology structure, its topology-aware score is the upper bound of all models. In contrast, the GCN and HGNN explicitly smooth the vertex with neighbors on the graph/hypergraph, which is the lower bound of all models. The proposed LightHGNNs achieve a trade-off score since the HGNNs is the teacher and the MLPs is the architecture. We notice that the LightHGNN$^+$ obtains a higher score than the LightHGNN, which indicates that learned vertex embeddings have lower topology relevance to the hypergraph. In the topology-aware distillation,

those reliable hyperedges are selected by the high resistance to the noise, which is more relevant to the task than those hyperedges with lower reliable scores. Thus, the LightHGNN$^+$ can resist the over-smoothing of features and topology, thus yielding better performance than LightHGNN. In Appendix K, we further provide visualization to demonstrate that the LightHGNN$^+$ will pull those vertices in hyperedges with higher reliable scores than those with lower scores.

## 5.5 ABLATION STUDIES

In this subsection, we conduct four ablation studies of the proposed LightHGNNs on the DBLP-Paper dataset as shown in Figure 4. We also provide extra ablation studies on the inductive ratio under production setting (Appendix I) and distillation with different teacher HGNNs (Appendix H).

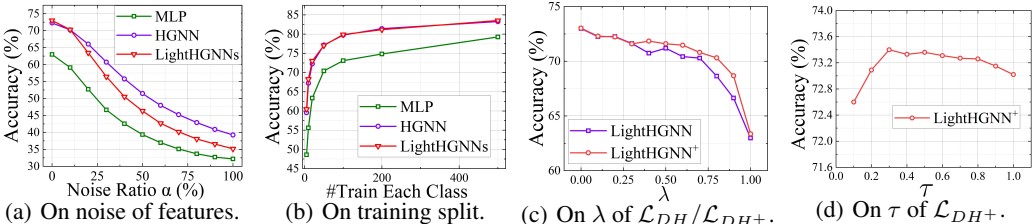

(a) On noise of features.    (b) On training split.    (c) On $\lambda$ of $\mathcal{L}_{DH}/\mathcal{L}_{DH+}$.    (d) On $\tau$ of $\mathcal{L}_{DH+}$.

Figure 4: Experimental results of ablation studies.

**On noise of vertex features.** We investigate the robustness by adding Gaussian noise on the vertex features $\boldsymbol{X}$: $\tilde{\boldsymbol{X}} = (1 - \alpha)\boldsymbol{X} + \alpha\epsilon$, where $\epsilon \sim \mathcal{N}(0, 0.01)$. The model is trained with the original feature $\boldsymbol{X}$ and evaluated with the noise feature $\tilde{\boldsymbol{X}}$. As shown in Figure 4(a), the LightHGNNs achieve the middle performance between MLPs and HGNN. When given a small noise ratio, the LightHGNNs is still better than HGNN. As the noise ratio rises, the performance of LightHGNNs is marching to that of MLPs. This is because the HGNN has an extra hypergraph structure as input, while MLPs and LightHGNNs rely only on the vertex features to make predictions. As the noise ratio rises, the input information will be submerged in the noise, which leads to worse performance.

**On training split under the transductive setting.** In Figure 4(b), we show the ablation study on the number of training samples under the transductive setting. The number of training samples for each class varies from 5 to 500. Obviously, LightHGNNs exhibits competitive performance with the HGNN and has significant advantages over MLPs. As for 5 training samples for each class, the margin between HGNN and MLPs is about 12%. Nevertheless, for 500 training samples for each class, the margin is only 4%. It indicates that the correlation information can supply the model better in the few information scenarios.

**Hyperparameter sensitivity on $\lambda$ and $\tau$.** In Figures 4(c) and 4(d), we provide the hyperparameter sensitivity of $\lambda$ and $\tau$ in the loss function of LightHGNNs. The $\lambda$ balances the weight of supervision from the true labels and soft labels, respectively. Due to the soft labels from teacher HGNN containing more information compared with the true labels Hinton et al. (2015), the performance of LightHGNNs decreases as the weight of the soft label decreases. However, the LightHGNN$^+$ can still outperform the LightHGNN via the extra high-order soft labels supervision. As for the temperature $\tau$, we find that too large or too small are both detrimental to the distillation. In practice, the $\tau = 0.5$ often yields pretty good performance, which is set in all datasets for a fair comparison.

## 6 CONCLUSION

In this paper, we propose LightHGNNs, including LightHGNN and LightHGNN$^+$, to bridge the gap between MLPs and HGNNs to achieve fast inference with low complexity. We design the reliable hyperedge quantification and sampling strategy to inject those task-relevant topology knowledge into the student MLPs. Extensive experiments on 11 real-world graph and hypergraph datasets indicate our LightHGNNs can achieve competitive performance to HGNNs and GNNs. Besides, experiments on a series of larger-scale synthetic hypergraph datasets indicate that by eliminating hypergraph dependency, our LightHGNNs can achieve $100\times$ faster inference, demonstrating the potential to deploy in realistic latency-constrained applications.

ACKNOWLEDGMENTS

This work was supported by National Natural Science Funds of China (No. 62088102, 62021002), Beijing Natural Science Foundation (No. 4222025).

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

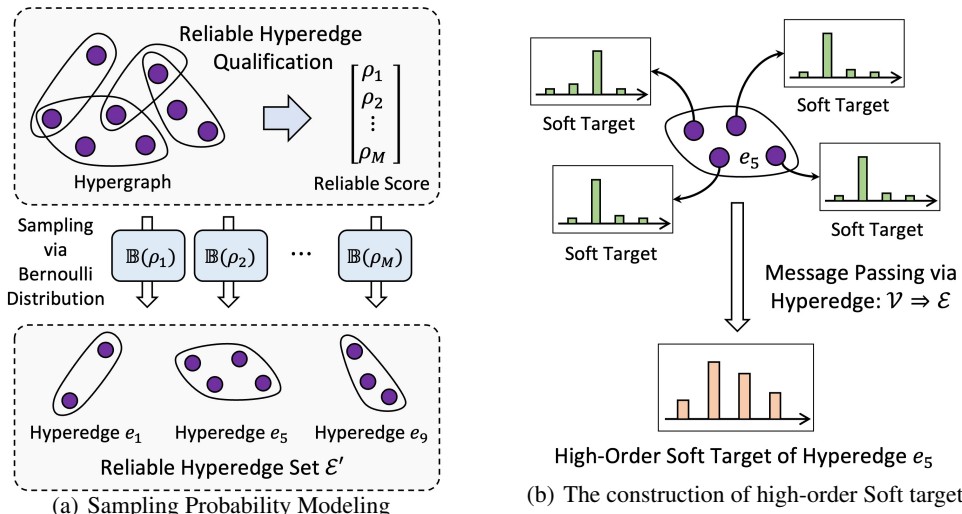

(a) Sampling Probability Modeling

(b) The construction of high-order Soft target.

Figure S1: The illustration of sampling probability modeling and high-order soft target.

## A  ALGORITHM

In this section, we provide the pseudo-code of the LightHGNN$^+$ framework for better understanding. As shown in Algorithm 1, to achieve HGNNs-to-MLPs knowledge distillation, we first pre-train the teacher HGNNs via the cross-entropy loss on the labeled vertices. Then, the soft targets and the hyperedge reliable scores are extracted based on the teacher HGNNs. In each epoch of the training process of the student MLPs, we sample the reliable hyperedge set from the total hyperedges and compute the high-order soft-label loss via those reliable hyperedges and the prediction of student and teacher. The original true label and soft-target supervision are also calculated for the final loss $\mathcal{L}_{hc}$ of LightHGNN$^+$. After training, the MLPs distilled from the HGNNs can be deployed for the online fast-inference environment without dependency on hypergraphs.

---

**Algorithm 1** Algorithm of LightHGNN$^+$ Framework.

---

**Input:** The hypergraph $\mathcal{G} = \{\mathcal{V}, \mathcal{E}\}$, Vertex feature matrix $\boldsymbol{X}$, Labeled dataset $\mathcal{D}^L = \{\mathcal{V}^L, \mathcal{Y}^L\}$. Number of epoch $E$.

**Output:** Predicted labels $\hat{\mathcal{Y}}^U$ for unlabeled vertices, Parameters of student MLPs $\{\boldsymbol{\Theta}\}_{l=1}^L$.

1: Pre-train the teacher HGNNs with label $\mathcal{Y}^L$.
2: Initialize the parameters $\{\boldsymbol{\Theta}\}_{l=1}^L$ of student MLPs.
3: $\{\hat{\boldsymbol{y}}_v^t \mid v \in \mathcal{V}\} \leftarrow$ extract the soft targets via teacher HGNNs.
4: $\{\rho_e \mid e \in \mathcal{E}\} \leftarrow$ reliable hyperedge qualification.
5: $t \leftarrow 0$
6: **while** $t < E$ **do**
7: $\quad t \leftarrow t + 1$
8: $\quad$ Predict the distribution of vertices $\{\hat{\boldsymbol{y}}_v^s \mid v \in \mathcal{V}\}$ via student MLPs.
9: $\quad$ Sample reliable hyperedge set $\mathcal{E}' = \{e \in \mathcal{E} \mid p(e) \sim \text{Bernoulli}(\rho_e)\}$ from the original hyperedge set $\mathcal{E}$.
10: $\quad$ Calculate the high-order soft targets $\{\hat{\boldsymbol{y}}_e^t \mid e \in \mathcal{E}'\}$ of hyperedges in the reliable hyperedge set via vertices' soft targets.
11: $\quad$ Calculate the student MLPs' prediction $\{\hat{\boldsymbol{z}}_e^s \mid e \in \mathcal{E}'\}$ of high-order soft targets of hyperedges in the reliable hyperedge set.
12: $\quad$ Calculate the total loss $\mathcal{L}_{DH+}$ of cross-entropy loss, soft-target loss, and high-order soft-target loss.
13: $\quad$ Update student MLPs' parameters $\{\boldsymbol{\Theta}\}_{l=1}^L$ by back propagation.
14: **end while**
15: **return** Predicted labels $\hat{\mathcal{Y}}^U$ for unlabeled vertices, Parameters of student MLPs $\{\boldsymbol{\Theta}\}_{l=1}^L$.

---

## B  TIME COMPLEXITY ANALYSIS

In this section, we provide the time complexity analysis during training and inference for deeper comparison, as shown in Table S1. In this table, the $L$ and $F$ denote the number of layers and the dimension of hidden layers, respectively. $N$ and $M$ are the number of vertices and hyperedges. $C$ indicates the number of categories. Clearly, the MLPs take $\mathcal{O}(LNF^2)$ time complexity during the training and inference. In comparison, the HGNN propagates messages via the $N \times N$ hypergraph Laplacian matrix, which needs extra $\mathcal{O}(LN^2F)$ time complexity. HGNN$^+$ further develops a two-stage message propagation vertex-hyperedge-vertex, which needs the $\mathcal{O}(LNMF)$ time complexity.

Table S1: The time complexity comparison during training and inference.

|  | MLPs | HGNN | HGNN$^+$ | LightHGNN | LightHGNN$^+$ |
|---|---|---|---|---|---|
| Training | $\mathcal{O}(LNF^2)$ | $\mathcal{O}(LN^2F + LNF^2)$ | $\mathcal{O}(LNMF + LNF^2)$ | $\mathcal{O}(LNF^2)$ | $\mathcal{O}(NMC + LNF^2)$ |
| Inference | $\mathcal{O}(LNF^2)$ | $\mathcal{O}(LN^2F + LNF^2)$ | $\mathcal{O}(LNMF + LNF^2)$ | $\mathcal{O}(LNF^2)$ | $\mathcal{O}(LNF^2)$ |

As for the proposed LightHGNN, it takes the same time complexity $\mathcal{O}(LN^2F)$ as the MLPs during the training and inference stage. During training, for better capture the high-order information, the LightHGNN$^+$ evaluates the reliable score of hyperedges and generates the high-order soft label, which both need to propagate the label distribution from vertex to hyperedge ($\mathcal{O}(NMC)$). Reliable hyperedge sampling only needs $\mathcal{O}(M)$. Therefore, the LightHGNN$^+$ takes $\mathcal{O}(NMC + LNF^2)$ time complexity for training and runs as fast as MLPs in inference.

## C  DATASET

We adopt three common-used graph datasets: Cora Sen et al. (2008), Pubmed McCallum et al. (2000), and Citeseer Giles et al. (1998), and eight hypergraph datasets: News20 Asuncion & Newman (2007), CA-Cora, CC-Cora, CC-Citeseer Yadati et al. (2019), DBLP-Paper, DBLP-Term, DBLP-Conf Sun et al. (2011), and IMDB-AW Fu et al. (2020). Table S2 provides the summary of these datasets. The Cora, Pubmed, and Citeseer are the paper citation networks, where each vertex denotes a scientific publication, and the label is the paper's topic. Each paper has a sparse bag-of-words feature vector and citation relationships among publications represented by corresponding edges. As for the hypergraph datasets, the CA-Cora, CC-Cora, and CC-Citeseer are also the publication datasets, where the vertices that are co-authored (CA) by an author or co-cited (CC) by a publication are connected in one hyperedge. The label of the vertex is also the topic of the publication. The authors are the vertices in the three hypergraphs: DBLP-Paper, DBLP-Term, and DBLP-Conf. The hyperedge constructed by the cooperating a paper, using the same term, published in the same conference, respectively. The label of the vertex is the research area of the author. As for the IMDB-AW dataset, the vertex is a movie, and the label is the corresponding category. The

Table S2: Statics of datasets. "#Edge" denotes the number of edges or hyperedges.

| Datasets | Type | #Nodes | #Edge | #Features | $\bar{d}_v$ | $\bar{d}_e$ | #Classes |
|---|---|---|---|---|---|---|---|
| *Cora* | Graph | 2,708 | 7,440 | 1,433 | 4.8 | 2 | 7 |
| *Pubmed* | Graph | 19,717 | 54,944 | 500 | 5.5 | 2 | 3 |
| *Citeseer* | Graph | 3,327 | 6,590 | 3,703 | 3.7 | 2 | 6 |
| *News20* | Hypergraph | 16,342 | 100 | 100 | 4.0 | 327.7 | 4 |
| *CA-Cora* | Hypergraph | 2,708 | 970 | 1,433 | 1.7 | 3.6 | 7 |
| *CC-Cora* | Hypergraph | 2,708 | 1,483 | 1,433 | 2.1 | 2.1 | 7 |
| *CC-Citeseer* | Hypergraph | 3,312 | 1,004 | 3,703 | 1.5 | 1.8 | 6 |
| *DBLP-paper* | Hypergraph | 4,057 | 5,701 | 334 | 2.3 | 1.6 | 4 |
| *DBLP-term* | Hypergraph | 4,057 | 6,089 | 334 | 28.6 | 19.1 | 4 |
| *DBLP-Conf* | Hypergraph | 4,057 | 20 | 334 | 4.8 | 982.2 | 4 |
| *IMDB-AW* | Hypergraph | 4,278 | 5,257 | 3,066 | 3.5 | 2.9 | 3 |

dataset contains two types of hyperedges: the co-actor and the co-writer relationships. For each actor or writer, participating in movies is connected by a hyperedge.

# D    TRANSDUCTIVE SETTING AND PRODUCTION SETTING

The transductive setting is widely adopted in the evaluation of the vertex classification on the graph and hypergraph datasets. In this setting, the vertex set $\mathcal{V}$ is divided into two sub-sets: labeled vertex set $\mathcal{V}^L$ and unlabeled vertex set $\mathcal{V}^U$. The unlabeled vertex set is used for testing and final evaluation. At the same time, the labeled vertex is further divided into the training set $\mathcal{V}^L_{tr}$ and validation set $\mathcal{V}^L_{va}$. In the training phase, a big hypergraph $\mathcal{G}$ including vertices from $\mathcal{V}^L_{tr} \cup \mathcal{V}^L_{va} \cup \mathcal{V}^L_{te}$ is constructed for message passing. However, only labels of those vertices from the training set $\mathcal{V}^L_{tr}$ are used to supervise the model's training, and the validation set $\mathcal{V}^L_{va}$ is adopted for "best model" selection. Finally, we report the performance of the "selected best model" on the testing set $\mathcal{V}^U$. Note that, in this setting, those vertices in the testing set $\mathcal{V}^U$ are still visible in the training phase. The labels of vertices from the validation set $\mathcal{V}^L_{va}$ and testing set $\mathcal{V}^L_{te}$ are absolutely unknown in the training phase. The detailed splitting of different datasets under the transductive setting is provided in Table S3.

Table S3: Splitting of datasets under the transductive setting.

| Datasets | #Nodes | #Classes | #Training | #Validation | #Testing |
|---|---|---|---|---|---|
| *Cora* | 2,708 | 7 | 140 | 700 | 1,868 |
| *Pubmed* | 19,717 | 3 | 60 | 300 | 19,357 |
| *Citeseer* | 3,327 | 6 | 120 | 600 | 2,607 |
| *News20* | 16,342 | 4 | 80 | 400 | 15,862 |
| *CA-Cora* | 2,708 | 7 | 140 | 700 | 1,868 |
| *CC-Cora* | 2,708 | 7 | 140 | 700 | 1,868 |
| *CC-Citeseer* | 3,312 | 6 | 120 | 600 | 2,592 |
| *DBLP-Paper* | 4,057 | 4 | 80 | 400 | 3,577 |
| *DBLP-Term* | 4,057 | 4 | 80 | 400 | 3,577 |
| *DBLP-Conf* | 4,057 | 4 | 80 | 400 | 3,577 |
| *IMDB-AW* | 4,278 | 3 | 60 | 300 | 3,918 |

However, the transductive setting is not the best way to evaluate a deployed model in the real world, where the unseen vertices usually appear in the testing set. Consequently, in this paper, we utilize the realistic production setting, which contains both transductive and inductive predictions. In the production setting, the unlabeled set $\mathcal{V}^U$ is further divided into transductive testing set $\mathcal{V}^U_t$ and inductive testing set $\mathcal{V}^U_i$. In the training phase, vertices from $\mathcal{V}_{obs} = \mathcal{V}^L_{tr} \cup \mathcal{V}^L_{va} \cup \mathcal{V}^U_t$ are utilized to construct a sub hypergraph $\mathcal{G}_{sub}$ for messages passing. The model training and selection are the same in the transductive setting. Then, we can fetch three types of performance, including under transductive setting, inductive setting, and production setting, respectively. The performance under the transductive and inductive settings is obtained by the big hypergraph $\mathcal{G}$ and the transductive testing set $\mathcal{V}^U_t$ as input and sub-hypergraph $\mathcal{G}_{sub}$ and inductive testing set $\mathcal{V}^U_i$ as input, respectively.

Table S4: Splitting of datasets under the production setting.

| Datasets | #Nodes | #Classes | #Training | #Validation | #Transductive Testing | #Inductive Testing |
|---|---|---|---|---|---|---|
| *News20* | 16,342 | 4 | 80 | 400 | 12,689 | 3,172 |
| *CA-Cora* | 2,708 | 7 | 140 | 700 | 1,494 | 373 |
| *CC-Cora* | 2,708 | 7 | 140 | 700 | 1,494 | 373 |
| *CC-Citeseer* | 3,312 | 6 | 120 | 600 | 2,073 | 518 |
| *DBLP-Paper* | 4,057 | 4 | 80 | 400 | 2,861 | 715 |
| *DBLP-Term* | 4,057 | 4 | 80 | 400 | 2,861 | 715 |
| *DBLP-Conf* | 4,057 | 4 | 80 | 400 | 2,861 | 715 |
| *IMDB-AW* | 4,278 | 3 | 60 | 300 | 3,134 | 783 |
| *Recipe-100k* | 101,585 | 8 | 160 | 800 | 80,500 | 20,125 |
| *Recipe-200k* | 240,094 | 8 | 160 | 800 | 191,307 | 47,826 |

The performance under the production setting is obtained by the big hypergraph $\mathcal{G}$ and the testing set $\mathcal{V}^U = \mathcal{V}_t^U \cup \mathcal{V}_i^U$ as input. In our experiments, 20% vertices of the testing set $\mathcal{V}^U$ are adopted for the inductive testing and the rest for the transductive testing. The detailed splitting of different datasets under the production setting is provided in Table S4.

## E   TRAINING DETAILS

We run 5 times with different random seeds for each experiment and report the average score and standard deviation. In each run, 20 samples from each category are selected for training, and 100 samples from each category are selected for validation. The rest is used for testing. The accuracy is adopted for performance comparison, and the model with the best performance in the validation set is applied to the test set for the results. Adam is adopted for optimization. KRD is implemented based on their released code [1]. The experiments of other baselines and our methods are implemented using Pytorch and DHG library [2]. As for the GNNs-to-MLPs methods, the teacher is the GCN, and the student is the MLPs. Besides, in the training phase, the same hyper-parameters are used for all methods to achieve a fair comparison, as shown in Tables S5 and S6. We run all experiments on a machine with 40 Intel(R) Xeon(R) E5-2640 v4 @2.40GHz CPUs, and a single NVIDIA GeForce RTX 3090 GPU.

Table S5: Hyper-parameter configuration of GCNs and HGNNs.

|  | GCN | GAT | HGNN | HGNN+ | UniGNN | UniGAT |
|---|---|---|---|---|---|---|
| Learning Rate | 0.01 | 0.01 | 0.01 | 0.01 | 0.01 | 0.01 |
| Weight Decay | 0.0005 | 0.0005 | 0.0005 | 0.0005 | 0.0005 | 0.0005 |
| Dropout Rate | 0.5 | 0.5 | 0.5 | 0.5 | 0.5 | 0.5 |
| Hidden Dimension | 32 | 8 | 32 | 32 | 32 | 8 |
| #Attention Heads | - | 4 | - | - | - | 4 |
| #Layer | 2 | 2 | 2 | 2 | 2 | 2 |

Table S6: Hyper-parameter configuration of GNNs-to-MLPs and LightHGNNs.

|  | MLPs | GLNN | KRD | LightHGNN | LightHGNN$^+$ |
|---|---|---|---|---|---|
| Learning Rate | 0.01 | 0.01 | 0.01 | 0.01 | 0.01 |
| Weight Decay | 0.0005 | 0.0005 | 0.0005 | 0.0005 | 0.0005 |
| Dropout Rate | 0.5 | 0.5 | 0.5 | 0.5 | 0.5 |
| Hidden Dimension | 128 | 128 | 128 | 128 | 128 |
| $\lambda$ | - | 0 | 0 | 0 | 0 |
| $\tau$ | - | - | 0.5 | 0.5 | 0.5 |
| #Layer | 2 | 2 | 2 | 2 | 2 |

**Multi-Layer Perceptrons (MLPs).**   To achieve efficient inference, the naive MLPs is adopted. The $l$-th layer of the MLPs is defined as:

$$z^{l+1} = \text{Dropout}\left(\sigma(z^l \Theta^l)\right), \tag{7}$$

where the $z^l$ is the embedding as the input of $l$-th Layer, and the $\sigma(\cdot)$ is a non-linear activation function. $\{\Theta^l\}_{l=1}^L$ is the learnable parameters of the MLPs. By default, the last layer removes the dropout and activation functions.

## F   GRAPH MODELS ON HYPERGRAPH DATASETS AND HYPERGRAPH MODELS ON GRAPH DATASETS

In this section, we introduce how to deploy the graph models on the hypergraph datasets and how to deploy the hypergraph models on the graph datasets. As for the graph datasets, if the method is a

---

[1] https://github.com/LirongWu/RKD

[2] https://github.com/iMoonLab/DeepHypergraph

hypergraph-based model like HGNN and HGNN$^+$, we will construct a hypergraph upon the original graph via the concatenation of the original pair-wise edge and the 1-hop neighborhood hyperedges as Feng et al. (2019). Given the graph, the 1-hop neighborhood hyperedges $\{e_i\}_{i=1}^N$ is the union of the hyperedge from each vertex's 1-hop neighborhood. For the vertex $v_i$, its 1-hop neighborhood hypergraph can be defined as:

$$e_i = \{v_i\} \cup \{v_j \mid v_j \in \mathcal{N}(v_i)\}, \tag{8}$$

where $\mathcal{N}(\cdot)$ represents the neighbor set of the specified vertex. By constructing the hypergraph from the graph in this way, the HGNNs can fully utilize the high-order representation and learning capability in the simple graph. As for the hypergraph datasets, if the method is a graph-based model like GCN and GAT, the clique expansion Gao et al. (2022) is utilized to transfer the hypergraph to the graph structure. Specifically, given hypergraph $\mathcal{G} = \{\mathcal{V}, \mathcal{E}\}$, for each hyperedge, we link every pair of vertices in the hyperedge to generate edges as follows:

$$\boldsymbol{A}(i, j) = \begin{cases} 1 & \text{if} \quad v_i, v_j \in e \quad \text{and} \quad e \in \mathcal{E} \\ 0 & \text{else} \end{cases}, \tag{9}$$

where $\boldsymbol{A} \in \{0, 1\}^{N \times N}$ is the adjacency matrix of the generated graph from the hypergraph.

## G   DETAILS OF HOW FAST ARE THE PROPOSED LIGHTHGNNS

To exploit the inference potential in larger hypergraph datasets, we manually synthesize a series of hypergraphs as shown in Table S7. The 12 synthetical hypergraphs are generated by the function 'dhg.random.hypergraph_Gnm()' of the DHG library. The number of vertices ($N$) varies from 1000 to 55000, and the number of hyperedges is fixed to $N/8$. The dimensions of vertex features and hidden layers are set to 16. Given the synthetic hypergraph, we report the inference time of the entire hypergraph and vertex feature matrix as input, like the transductive setting. Obviously, as the scale of the hypergraph increases, the inference time of HGNNs increases exponentially. In contrast, the LightHGNNs still exhibit stable inference speed. The advantage of LightHGNNs inference speed increases as the size of the hypergraph grows.

In Table S8, we conduct experiments on two larger hypergraph datasets: Recipe-100k (10w vertices) and Recipe-200k (24w vertices). As shown in the table, our LightHGNNs can not only achieve better performance but also extremely reduce the inference time. As the scale of the dataset increases, the advantage of our LightHGNNs becomes more evident. Naive HGNN becomes slower and slower, even throwing the Out-Of-Memory (OOM) error confronting 10w+ vertices, while our LightHGNNs are still fast. Besides, our LightHGNNs exhibit better performance compared to the naive HGNN under the transductive testing set (about 1w or 2w vertices for testing), which demonstrates the effectiveness of our distillation framework. Besides, we provide the accuracy, memory, and runtime comparison of methods with different configurations on the IMDB-AW dataset in Table S9.

Table S7: Inference time ($ms$) comparison on 12 synthetic hypergraphs.

| #Vertices of Hypergraph | HGNN | HGNN$^+$ | LightHGNNs | Faster |
|---|---|---|---|---|
| 1,000 | 0.47 | 0.98 | 0.39 | 1.2× |
| 2,000 | 0.50 | 0.98 | 0.40 | 1.3× |
| 4,000 | 1.10 | 1.02 | 0.42 | 2.6× |
| 10,000 | 5.39 | 3.77 | 0.44 | 12.3× |
| 20,000 | 19.00 | 14.79 | 0.56 | 33.9× |
| 25,000 | 29.78 | 22.92 | 0.67 | 44.4× |
| 30,000 | 42.44 | 29.60 | 0.78 | 54.4× |
| 35,000 | 56.85 | 45.04 | 0.91 | 62.5× |
| 40,000 | 73.72 | 57.18 | 1.03 | 71.6× |
| 45,000 | 92.30 | 71.98 | 1.15 | 80.3× |
| 50,000 | 113.20 | 88.42 | 1.26 | 89.8× |
| 55,000 | 137.70 | 107.24 | 1.37 | 100.5× |

Table S8: Experimental results on two larger-scale hypergraph datasets. "#Testing" indicates "The number of vertices for testing". "#Infer" denotes the "Inference Time". "OOM" denotes the "Out of Memory" error.

| | | Recipe-100k | | | Recipe-200k | | |
|---|---|---|---|---|---|---|---|
| | | #Testing | Accuracy | #Infer | #Testing | Accuracy | #Infer |
| HGNN | Trans. | 10,063 | $41.26_{\pm 3.08}$ | 5.39 | 23,914 | $37.25_{\pm 4.88}$ | 29.78 |
| | Ind. | 90,562 | OOM | $\infty$ | 215,220 | OOM | $\infty$ |
| | Prod. | 100,625 | OOM | $\infty$ | 239,134 | OOM | $\infty$ |
| LightHGNN | Trans. | 10,063 | $41.82_{\pm 3.48}$ | 0.44 | 23,914 | $38.48_{\pm 5.59}$ | 0.67 |
| | Ind. | 90,562 | $41.15_{\pm 3.23}$ | 2.18 | 215,220 | $37.62_{\pm 5.89}$ | 5.05 |
| | Prod. | 100,625 | $41.22_{\pm 3.26}$ | 2.40 | 239,134 | $37.70_{\pm 5.86}$ | 5.62 |
| LightHGNN$^+$ | Trans. | 10,063 | $42.50_{\pm 3.74}$ | 0.44 | 23,914 | $38.76_{\pm 5.24}$ | 0.67 |
| | Ind. | 90,562 | $42.28_{\pm 3.62}$ | 2.18 | 215,220 | $38.26_{\pm 5.67}$ | 5.05 |
| | Prod. | 100,625 | $42.31_{\pm 3.63}$ | 2.40 | 239,134 | $38.31_{\pm 5.62}$ | 5.62 |

Table S9: The comparison of accuracy, memory, and runtime on the IMDB-AW dataset.

| Methods | Runtime (ms) | Memory (MB) | Accuracy (%) |
|---|---|---|---|
| MLPs-L1 | 0.26 | 4.27 | $36.48_{\pm 1.87}$ |
| MLPs-L3 | 0.52 | 12.63 | $40.89_{\pm 1.13}$ |
| MLPs-w32 | 0.35 | 8.45 | $40.00_{\pm 1.38}$ |
| MLPs-w64 | 0.40 | 16.80 | $40.46_{\pm 1.82}$ |
| MLPs | 0.40 | 33.51 | $40.77_{\pm 1.15}$ |
| MLPs-w256 | 0.53 | 66.94 | $41.63_{\pm 1.28}$ |
| HGNN-L1 | 0.35 | 144.94 | $48.58_{\pm 1.23}$ |
| HGNN | 0.53 | 289.79 | $50.78_{\pm 1.67}$ |
| HGNN-L3 | 0.65 | 434.64 | $49.01_{\pm 2.25}$ |
| HGNN-w64 | 0.57 | 300.24 | $50.44_{\pm 1.76}$ |
| HGNN-w128 | 0.83 | 321.13 | $49.77_{\pm 1.63}$ |
| HGNN-w256 | 1.45 | 362.90 | $49.63_{\pm 1.95}$ |
| HGNN$^+$-L1 | 0.77 | 6.60 | $50.78_{\pm 1.67}$ |
| HGNN$^+$ | 1.31 | 13.10 | $50.77_{\pm 1.75}$ |
| HGNN$^+$-L3 | 1.88 | 19.61 | $48.48_{\pm 1.40}$ |
| HGNN$^+$-w64 | 1.53 | 26.12 | $50.37_{\pm 1.99}$ |
| HGNN$^+$-w128 | 2.12 | 52.14 | $50.40_{\pm 1.81}$ |
| HGNN$^+$-w256 | 3.35 | 104.18 | $50.06_{\pm 1.73}$ |
| LightHGNNs-w32 | 0.35 | 8.45 | $46.60_{\pm 7.16}$ |
| LightHGNNs-w64 | 0.40 | 16.80 | $50.77_{\pm 1.61}$ |
| LightHGNNs | 0.40 | 33.51 | $51.53_{\pm 1.67}$ |
| LightHGNNs-w256 | 0.53 | 66.94 | $50.32_{\pm 2.68}$ |

## H  ABLATION STUDY ON TEACHER HGNNS ARCHITECTURE

In this section, we further investigate the performance of the LightHGNNs with different teacher architectures on the DBLP-Paper dataset, as shown in Table S10. Among these methods, HGNN Feng et al. (2019) serves as our baseline approach. HGNN$^+$ Gao et al. (2022) extends HGNN by introducing a general framework for modeling high-order data correlations. UniGNN, UniGAT, and UniGCNII Huang & Yang (2021) generalize pre-designed graph neural network architectures to hypergraphs. ED-HNN Wang et al. (2023) combines the star expansions of hypergraphs with standard message passing neural networks. AllSet Chien et al. (2022) implements hypergraph neural network layers as compositions of two multiset functions that can be efficiently learned for each task and each dataset. The "Original" in the table denotes the performance of the teacher architecture. We observe that the LightHGNNs show robust competitive performance compared with different teachers. Experimental results indicate that our method can adapt to different teachers, and the

better the teacher network's performance, the better our method's performance will be improved accordingly.

Table S10: Experimental results of LightHGNNs with different teacher Architectures.

| Method | Setting | Original | LightHGNN | LightHGNN$^+$ |
|---|---|---|---|---|
| HGNN | Prod. | $\underline{71.52}_{\pm 1.31}$ | $71.14_{\pm 1.23}$ | $\mathbf{71.69}_{\pm 1.44}$ |
| | Tran. | $70.75_{\pm 1.49}$ | $\underline{70.88}_{\pm 1.29}$ | $\mathbf{71.40}_{\pm 1.50}$ |
| | Ind. | $\underline{72.72}_{\pm 2.32}$ | $72.22_{\pm 2.08}$ | $\mathbf{72.86}_{\pm 2.33}$ |
| HGNN$^+$ | Prod. | $\mathbf{72.91}_{\pm 1.23}$ | $72.63_{\pm 0.80}$ | $\underline{72.65}_{\pm 0.66}$ |
| | Tran. | $71.83_{\pm 1.50}$ | $\underline{72.33}_{\pm 0.89}$ | $\mathbf{72.47}_{\pm 0.95}$ |
| | Ind. | $\mathbf{74.43}_{\pm 2.39}$ | $73.84_{\pm 1.69}$ | $\underline{74.15}_{\pm 1.66}$ |
| UniGNN | Prod. | $\underline{72.73}_{\pm 1.45}$ | $72.66_{\pm 1.48}$ | $\mathbf{72.83}_{\pm 1.66}$ |
| | Tran. | $71.88_{\pm 1.51}$ | $\underline{72.38}_{\pm 1.54}$ | $\mathbf{72.40}_{\pm 1.66}$ |
| | Ind. | $73.79_{\pm 2.57}$ | $\underline{73.79}_{\pm 1.68}$ | $\mathbf{74.54}_{\pm 2.58}$ |
| UniGAT | Prod. | $\mathbf{73.00}_{\pm 1.40}$ | $\underline{72.74}_{\pm 1.20}$ | $72.59_{\pm 1.32}$ |
| | Tran. | $71.99_{\pm 1.49}$ | $\mathbf{72.46}_{\pm 1.36}$ | $\underline{72.32}_{\pm 1.21}$ |
| | Ind. | $\mathbf{74.29}_{\pm 2.53}$ | $\underline{73.90}_{\pm 1.69}$ | $73.65_{\pm 2.64}$ |
| ED-HNN | Prod. | $72.01_{\pm 1.06}$ | $\underline{73.45}_{\pm 1.48}$ | $\mathbf{73.53}_{\pm 1.61}$ |
| | Tran. | $70.64_{\pm 1.52}$ | $\underline{72.11}_{\pm 1.66}$ | $\mathbf{73.34}_{\pm 1.78}$ |
| | Ind. | $73.79_{\pm 1.72}$ | $\mathbf{74.79}_{\pm 1.61}$ | $\underline{74.32}_{\pm 1.86}$ |
| UniGCNII | Prod. | $72.94_{\pm 0.97}$ | $\underline{73.29}_{\pm 1.29}$ | $\mathbf{73.58}_{\pm 1.69}$ |
| | Tran. | $72.04_{\pm 0.84}$ | $\underline{72.87}_{\pm 1.12}$ | $\mathbf{73.05}_{\pm 1.60}$ |
| | Ind. | $73.81_{\pm 2.48}$ | $\underline{74.95}_{\pm 2.30}$ | $\mathbf{75.71}_{\pm 2.54}$ |
| AllSet | Prod. | $71.37_{\pm 1.60}$ | $\underline{73.20}_{\pm 1.96}$ | $\mathbf{73.53}_{\pm 1.48}$ |
| | Tran. | $71.15_{\pm 1.93}$ | $\underline{73.00}_{\pm 2.57}$ | $\mathbf{73.14}_{\pm 1.32}$ |
| | Ind. | $72.97_{\pm 2.61}$ | $\mathbf{75.36}_{\pm 2.07}$ | $\underline{75.10}_{\pm 2.61}$ |
| Avg. Rank | | 2.43 | 2.1 | 1.43 |

## I  ABLATION STUDY ON INDUCTIVE RATIO UNDER PRODUCTION SETTING

In this section, we adjust the ratio of the inductive testing from $10\%$ to $90\%$ to investigate the influence of the unseen vertices ratio on the performance in the DBLP-Paper dataset. Experimental results are shown in Table S11. Based on the results, we have three observations. Firstly, as the ratio of inductive testing increases, the performance of the three types of settings all show a slight decline. This is because a task under the inductive setting is inherently harder than that under the transductive setting due to the seen topology being less. Secondly, we observe that the proposed LightHGNNs often show competitive performance to the HGNNs and exhibit significant improvement to the MLPs, demonstrating the proposed methods' effectiveness. Thirdly, we notice that LighHGNN$^+$ shows better performance when the inductive testing ratio is small. This is because as the inductive testing ratio increases, the unseen hyperedges and vertex features also increase. This information, especially the unseen topology, will be seen in the inductive testing of the HGNNs, while it still is unseen in the testing of the LightHGNNs. As the unseen information increases, the performance of our LightHGNN will decrease accordingly.

## J  MORE EVALUATION ON THE NUMBER OF LAYERS AND THE TOPOLOGY-AWARE SCORE.

In Section 5.4, we attempt to define a specific metric to quantify the over-smoothing phenomenon and design a topology-aware score $\mathcal{S}$ to measure the relevance of features and hypergraph typology. Specifically, we first calculate the cosine similarity between the vertex and hyperedge features and then calculate the average cosine similarity of all vertices in the hypergraph. The topology-aware score $\mathcal{S}$ is defined as the average cosine similarity of all hyperedges in the hypergraph. The higher

Table S11: Experimental results of ablation study on inductive ratio under production setting.

| Ind./Trans. Ratio | Setting | MLP | HGNN | LightHGNN | LightHGNN$^+$ |
|---|---|---|---|---|---|
| 10%/90% | Prod. | $63.56_{\pm1.22}$ | $\mathbf{71.30}_{\pm1.82}$ | $70.91_{\pm1.35}$ | $\underline{71.06}_{\pm2.08}$ |
|  | Tran. | $63.49_{\pm1.40}$ | $\underline{70.95}_{\pm2.14}$ | $70.76_{\pm1.37}$ | $\mathbf{70.98}_{\pm2.24}$ |
|  | Ind. | $64.20_{\pm1.92}$ | $\underline{72.04}_{\pm2.53}$ | $\mathbf{72.26}_{\pm2.65}$ | $71.82_{\pm2.00}$ |
| 20%/80% | Prod. | $63.56_{\pm1.15}$ | $\underline{71.52}_{\pm1.31}$ | $71.14_{\pm1.23}$ | $\mathbf{71.69}_{\pm1.44}$ |
|  | Tran. | $63.37_{\pm1.17}$ | $70.75_{\pm1.49}$ | $\underline{70.88}_{\pm1.29}$ | $\mathbf{71.40}_{\pm1.50}$ |
|  | Ind. | $64.30_{\pm1.50}$ | $\underline{72.72}_{\pm2.32}$ | $72.22_{\pm2.08}$ | $\mathbf{72.86}_{\pm2.33}$ |
| 30%/70% | Prod. | $63.36_{\pm1.52}$ | $71.58_{\pm0.73}$ | $\underline{71.69}_{\pm0.91}$ | $\mathbf{72.08}_{\pm0.70}$ |
|  | Tran. | $63.10_{\pm1.30}$ | $70.31_{\pm1.36}$ | $\underline{71.21}_{\pm1.30}$ | $\mathbf{71.64}_{\pm1.18}$ |
|  | Ind. | $63.97_{\pm2.37}$ | $72.61_{\pm1.24}$ | $\underline{72.80}_{\pm0.89}$ | $\mathbf{73.10}_{\pm0.73}$ |
| 40%/60% | Prod. | $63.29_{\pm1.22}$ | $70.44_{\pm1.61}$ | $\underline{70.73}_{\pm1.75}$ | $\mathbf{70.93}_{\pm1.38}$ |
|  | Tran. | $62.87_{\pm1.23}$ | $68.92_{\pm2.20}$ | $\underline{70.41}_{\pm2.07}$ | $\mathbf{70.52}_{\pm1.69}$ |
|  | Ind. | $63.93_{\pm1.46}$ | $71.03_{\pm1.20}$ | $\underline{71.20}_{\pm1.30}$ | $\mathbf{71.55}_{\pm0.98}$ |
| 50%/50% | Prod. | $63.17_{\pm1.56}$ | $\mathbf{70.84}_{\pm0.79}$ | $\underline{70.57}_{\pm0.61}$ | $70.31_{\pm0.60}$ |
|  | Tran. | $63.23_{\pm1.66}$ | $69.21_{\pm1.01}$ | $\mathbf{70.82}_{\pm1.48}$ | $\underline{70.48}_{\pm0.88}$ |
|  | Ind. | $63.11_{\pm1.59}$ | $\mathbf{70.76}_{\pm0.71}$ | $\underline{70.33}_{\pm0.50}$ | $70.14_{\pm1.19}$ |
| 60%/40% | Prod. | $63.32_{\pm1.30}$ | $\mathbf{70.24}_{\pm0.73}$ | $69.27_{\pm0.69}$ | $\underline{70.15}_{\pm0.70}$ |
|  | Tran. | $63.15_{\pm1.67}$ | $68.88_{\pm0.98}$ | $\underline{69.53}_{\pm0.99}$ | $\mathbf{70.30}_{\pm1.36}$ |
|  | Ind. | $63.43_{\pm1.28}$ | $\mathbf{70.11}_{\pm0.98}$ | $69.09_{\pm1.38}$ | $\underline{70.06}_{\pm1.23}$ |
| 70%/30% | Prod. | $63.28_{\pm1.56}$ | $\mathbf{70.50}_{\pm1.40}$ | $69.93_{\pm1.61}$ | $\underline{69.98}_{\pm1.42}$ |
|  | Tran. | $63.25_{\pm2.35}$ | $68.79_{\pm1.68}$ | $\underline{69.79}_{\pm2.10}$ | $\mathbf{69.98}_{\pm2.11}$ |
|  | Ind. | $63.30_{\pm1.53}$ | $\mathbf{70.41}_{\pm1.25}$ | $69.98_{\pm1.53}$ | $\underline{69.98}_{\pm1.34}$ |
| 80%/20% | Prod. | $63.24_{\pm1.62}$ | $\mathbf{70.12}_{\pm1.45}$ | $\underline{68.98}_{\pm1.28}$ | $68.84_{\pm1.11}$ |
|  | Tran. | $62.79_{\pm2.20}$ | $67.51_{\pm1.84}$ | $\mathbf{68.65}_{\pm2.16}$ | $\underline{68.18}_{\pm2.19}$ |
|  | Ind. | $63.36_{\pm1.57}$ | $\mathbf{70.14}_{\pm1.47}$ | $\underline{69.06}_{\pm1.37}$ | $69.01_{\pm1.24}$ |
| 90%/10% | Prod. | $63.38_{\pm1.51}$ | $\mathbf{69.81}_{\pm1.28}$ | $\underline{69.11}_{\pm0.99}$ | $68.34_{\pm1.47}$ |
|  | Tran. | $63.96_{\pm2.76}$ | $\underline{68.88}_{\pm2.25}$ | $\mathbf{69.94}_{\pm3.12}$ | $68.43_{\pm3.20}$ |
|  | Ind. | $63.32_{\pm1.53}$ | $\mathbf{69.66}_{\pm1.26}$ | $\underline{69.02}_{\pm1.02}$ | $68.33_{\pm1.42}$ |

the score, the more relevant the features and hypergraph typology. We first provide the accuracy of MLP, HGNN, and LightHGNNs with respect to different numbers of layers, as shown in Table S12. In the table, the accuracy of MLP, HGNN, and LightHGNNs all significantly decrease as the number of layers increases, especially for the MLP and HGNN. However, the accuracy of LightHGNN$^+$ is more stable than that of MLP, HGNN, and LightHGNNs. This is because the proposed LightHGNN$^+$ can alleviate the over-smoothing phenomenon via the hyperedge reliability sampling, thus yielding robust performance that combats the increase in the number of layers.

Table S12: Experimental results of different number of layers on IMDB-AW dataset.

| #Layer | MLP | HGNN | LightHGNN | LightHGNN$^+$ |
|---|---|---|---|---|
| 2 | $40.87_{\pm1.43}$ | $50.78_{\pm1.67}$ | $50.19_{\pm1.56}$ | $50.47_{\pm1.92}$ |
| 3 | $41.25_{\pm1.03}$ | $50.32_{\pm1.23}$ | $49.52_{\pm2.38}$ | $50.38_{\pm2.23}$ |
| 4 | $41.44_{\pm1.45}$ | $44.97_{\pm3.72}$ | $47.58_{\pm5.82}$ | $49.75_{\pm2.91}$ |
| 5 | $41.52_{\pm0.67}$ | $42.59_{\pm2.45}$ | $48.55_{\pm5.59}$ | $47.97_{\pm6.08}$ |
| 6 | $41.18_{\pm0.93}$ | $39.72_{\pm3.76}$ | $47.61_{\pm5.95}$ | $47.66_{\pm5.84}$ |
| 7 | $40.23_{\pm1.48}$ | $31.47_{\pm5.61}$ | $46.33_{\pm8.06}$ | $47.67_{\pm5.97}$ |
| 8 | $38.75_{\pm1.95}$ | $32.89_{\pm4.11}$ | $46.64_{\pm7.89}$ | $47.16_{\pm6.31}$ |
| 9 | $36.81_{\pm2.22}$ | $32.14_{\pm4.59}$ | $46.24_{\pm5.50}$ | $47.69_{\pm6.03}$ |
| 10 | $36.75_{\pm2.35}$ | $32.91_{\pm0.89}$ | $42.64_{\pm6.11}$ | $44.18_{\pm5.99}$ |

We further provide the topology-aware score $\mathcal{S}$ of MLP, HGNN, and LightHGNNs with respect to different numbers of layers, as shown in Table S13. Based on the results in the table, we have the following three observations. First, despite the increase in the number of layers, the topology-aware score of MLP is always higher than that of HGNN and LightHGNNs. This is because the MLP only relies on the vertex features and ignores the hypergraph typology. Second, the topology-aware score of LightHGNN$^+$ is more stable than MLP, HGNN, and LightHGNNs. This is because the proposed LightHGNN$^+$ can alleviate the over-smoothing phenomenon via the hyperedge reliability sampling, thus yielding robust performance that combats the increase in the number of layers. Third, we find an interesting phenomenon that the topology-aware score significantly decreases in #layer $6 \to 7$. The results explain when the accuracy of HGNN and LightHGNN significantly decreases in #layer $6 \to 7$, as shown in the above table. This is because the phenomenon of over-smoothing has reached a critical point with the increase in the number of layers, impacting the representations for hypergraphs. As a result, the performance has sharply declined. However, why the critical point is reached at this specific layer needs further investigation in future work. We believe this is an intriguing experimental result that will impact the research community's study of over-smoothing.

Table S13: Results of topology-aware scores on the IMDB-AW dataset.

| #Layer | MLP | HGNN | LightHGNN | LightHGNN$^+$ |
|---|---|---|---|---|
| 2 | 1.180 | 0.040 | 0.713 | 0.770 |
| 3 | 2.498 | 0.049 | 0.876 | 0.954 |
| 4 | 2.660 | 0.017 | 0.782 | 0.821 |
| 5 | 2.778 | 0.023 | 0.533 | 0.648 |
| 6 | 2.887 | 0.021 | 0.454 | 0.570 |
| 7 | 1.922 | 0.008 | 0.076 | 0.376 |
| 8 | 2.842 | 0.004 | 0.082 | 0.226 |
| 9 | 2.480 | 0.007 | 0.058 | 0.178 |
| 10 | 2.096 | 0.006 | 0.014 | 0.120 |

## K  VISUALIZATION

In this section, we provide the visualization of the hyperedge reliable score on different datasets as shown in Figure S2. To investigate whether the topology-aware distillation can adaptively inject the high-order information into the student, we calculate the distance (KL Divergence) of the corresponding hyperedge between the student LightHGNN$^+$ and the teacher HGNN. The x-coordinate

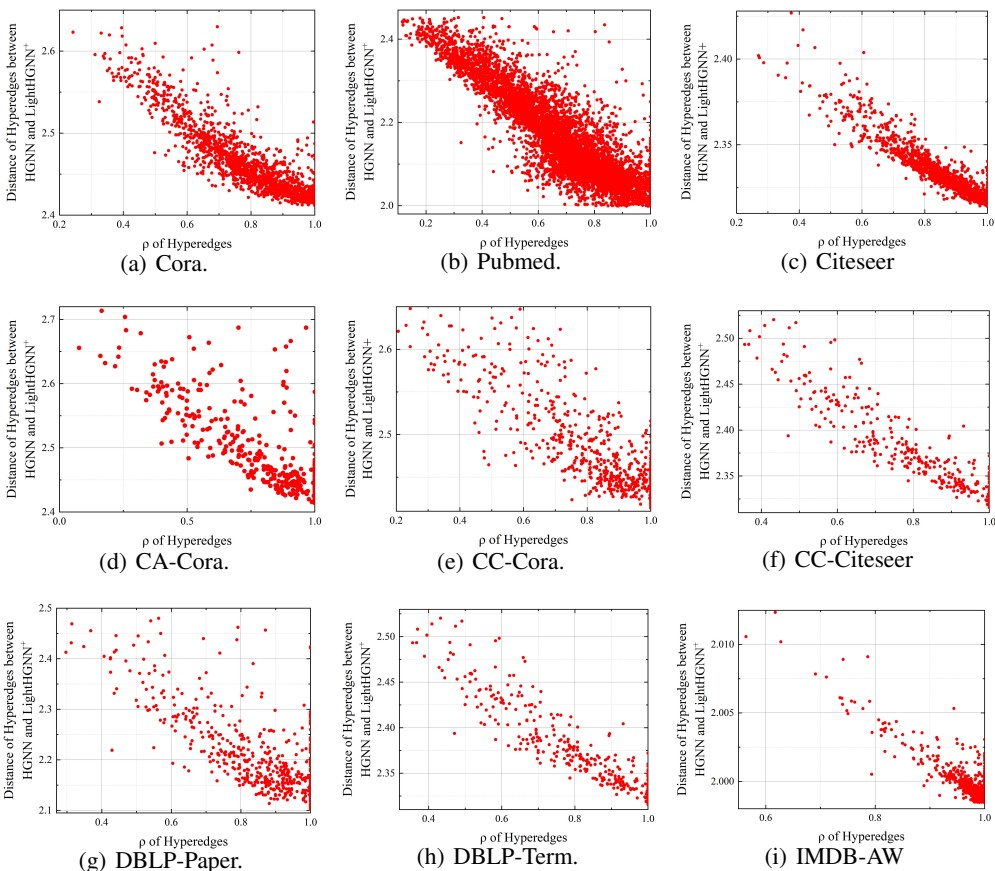

Figure S2: Visualization on the hyperedge reliable score.

denotes the hyperedge reliable score $\rho$ of each hyperedge, and the y-coordinate is the distance of the high-order soft targets of the teacher and student calculated by equation 3. Obviously, those hyperedges with higher reliable scores will be closer to the teacher, which demonstrates that our topology-aware distillation can adaptively inject reliable high-order information into the student. The teacher HGNNs blindly smooth vertex features via all hyperedges, while the student only focuses on a few reliable hyperedges and uses them to guide the message prorogation. This is also the main reason why our LightHGNN$^+$ can effectively resist over-smoothing, as stated in Section 5.4.

