# OpenReview forum: "LightHGNN: Distilling Hypergraph Neural Networks into MLPs for 100x Faster Inference"
_ICLR.cc/2024/Conference — ICLR 2024 poster_

### Official Review · Reviewer_U1AB · 2023-10-17

**Soundness:** 2 fair
**Presentation:** 2 fair
**Contribution:** 2 fair
**Rating:** 6
**Confidence:** 3

**Summary:**

This paper extends graph knowledge distillation to hypergraphs and achieves significant improvements in both efficiency and effectiveness. In particular, experiments on synthetic hypergraphs indicate LightHGNNs can run 100× faster than HGNNs, showing their ability for latency-sensitive deployments.

**Strengths:**

- This is the first work to explore the knowledge distillation on hypergraphs.
- The experiments in the paper are adequate, especially the results on the inference efficiency.
- The paper is well-written and presented.

**Weaknesses:**

- Lack of comparison with SOTA GNN-to-MLP KD methods such as NOSMOG [1].
- From the results in Tables 1 and 2, it seems that LightHGNN only achieves comparable rather than (significantly) better results than HGNN. Can the authors explain more about this?
- I read subsections 4.2.1 and 4.2.2 carefully and realized that the reliability quantification and sampling (which I understand to be the core design of this paper) seem to be very similar to RKD. In my opinion, it's okay to extend designs of previous work to hypergraphs, but it's better to add proper citations and clarify which parts are similar and which parts are the main contributions of this paper.
- Most of the datasets used in this paper are small, and the authors are encouraged to demonstrate effectiveness on more large-scale datasets.
- Can the authors explain more about how the proposed method captures high-order correlations? I think more discussions on the differences between hypergraph KD and general graph KD can greatly enhance the readability and contribution of the paper.

[1] Tian Y, Zhang C, Guo Z, et al. Learning mlps on graphs: A unified view of effectiveness, robustness, and efficiency[C]//The Eleventh International Conference on Learning Representations. 2023.

**Questions:**

Can the author answer the question I posed in the weakness part?

**Details Of Ethics Concerns:**

No ethics review is needed.

---

> ### Author Response · Authors · 2023-11-19
> **Response to Reviewer U1AB (Part 1/3)**
>
> ### **Response to W1**
> Thanks for your suggestions. We further add the experiments of NOSMOG[1] in the table of "Table 3: Experimental results on graph and hypergraph datasets.". A part of the results are shown in the following table.
>
> | Type | Model | Graph Dataset |  |  | Hypergraph Dataset |  |  |
> |:---:|:---:|:---:|:---:|:---:|:---:|:---:|:---:|
> |  |  | Cora | Pubmed | Citeseer | CA-Cora | DBLP-Paper | IMDB-AW |
> | GNNs-to-MLPs | GLNN | **80.93**$_{\pm 1.90}$ | 78.36$_{\pm 1.99}$ | 69.88$_{\pm 1.66}$ | 72.19$_{\pm 3.83}$ | 72.50$_{\pm 1.62}$ | 50.48$_{\pm 1.51}$ |
> |  | KRD | 79.47$_{\pm 1.73}$ | 78.72$_{\pm 1.94}$ | 69.82$_{\pm 3.36}$ | 71.75$_{\pm 3.53}$ | 72.85$_{\pm 6.76}$ | 49.65$_{\pm 2.12}$ |
> |  | NOSMOG[1] | 80.12$_{\pm 0.91}$ | **80.42**$_{\pm 0.33}$ | **70.86**$_{\pm 3.53}$ | 68.96$_{\pm 7.34}$ | 71.47$_{\pm 2.13}$ | 48.96$_{\pm 1.43}$ |
> | HGNNs-to-MLPs | LightHGNN | 80.36$_{\pm 2.06}$ | 79.15$_{\pm 1.57}$ | 69.17$_{\pm 3.27}$ | **73.63**$_{\pm 5.14}$ | 72.74$_{\pm 1.07}$ | 50.19$_{\pm 1.56}$ |
> |  | LightHGNN+ | 80.68$_{\pm 1.74}$ | 79.16$_{\pm 1.37}$ | 70.34$_{\pm 1.95}$ | 73.54$_{\pm 3.80}$ | **72.93**$_{\pm 0.97}$ | **50.78**$_{\pm 1.92}$ |
>
> As shown in the table, our LightHGNNs achieve comparable performance to NOSMOG[1] on graph datasets and significantly better performance on hypergraph datasets. The results demonstrate the effectiveness of our distillation framework confronting both graph and hypergraph datasets.
>
> [1] Tian et al. Learning mlps on graphs: A unified view of effectiveness, robustness, and efficiency. ICLR, 2023.
>
>
> ### **Response to W2**
> Thanks for your comments. We attribute the performance improvement of LightHGNNs to the following two reasons. First, knowledge distillation from teacher HGNN determines the student LightHGNNs can achieve comparable performance to HGNN. Second, the proposed hyperedge reliability sampling can help the student MLPs distinguish those noise hyperedges in the original hypergraph, which can help the student LightHGNN$^+$ capture more task-related and reliable information, alleviate over-smoothing (as empirically demonstrated in Section 5.4), thus yielding better performance.
>
>
> ### **Response to W3**
> Thanks for your suggestions. We will future clarify the difference between our work and KRD[1] and add proper citations in the revised version. The main difference between our work and KRD[1] is two-fold.
> - KRD [1] is a typical reliable-based distillation for graph neural networks. However, its student MLPs can only be supervised by the soft label from those reliable vertices (knowledge points in their paper), which still lose the structure information and cannot be utilized in the hypergraph. However, our LightHGNN$^+$ can quantify the reliability of the high-order correlations and inject the reliable high-order information into the student MLPs via explicit supervision from high-order soft labels of those reliable hyperedges. The proposed LightHGNN$^+$ can sufficiently take advantage of both the vertex features and high-order structure information, thus yielding better performance.
> - Besides, we design a metric to measure the over-smoothing phenomenon (the relevance of features and hypergraph typology) in hypergraph neural networks and empirically demonstrate that our LightHGNN$^+$ has the capability to combat the over-smoothing and achieve better performance, which is not considered in KRD[1].
>
> All those discussions will be added in the revised version.
>
> [1] Zhang et al. Graph-less neural networks: Teaching old MLPs new tricks via distillation. ICLR, 2022.

---

> ### Author Response · Authors · 2023-11-19
> **Response to Reviewer U1AB (Part 2/3)**
>
> ### **Response to W4**
> Thanks for your suggestions. We further conduct experiments to demonstrate the capability of our LightHGNNs on large-scale hypergraph inference. Two large scale hypergraph datasets[1]: Recipe-100k (10w vertices) and Recipe-200k (24w vertices) are employed here. The two datasets both contain eight categories. Specifically, the Recipe-100k dataset contains 101,585 vertices and 12,387 hyperedges, and the Recipe-200k dataset contains 240,094 vertices and 18,129 hyperedges. The vertex features are the bag of words from the sentence that makes the recipe. Hyperedges are the ingredients of the recipe or the Keywords for food preparation steps. Experiments on the two datasets are shown in the following table.
>
> |  |  | Recipe-100k |  |  | Recipe-200k |  |  |
> |---|---|:---:|:---:|:---:|:---:|:---:|:---:|
> |  |  | #Vertices of Testing | Acc.(%) | Infer Time(s) | #Vertices of Testing | Acc.(%) | Infer Time(s) |
> | HGNN | Trans. | 10,063 | 41.26$_{\pm 3.08}$ | 5.39 | 23,914 |  37.25$_{\pm 4.88}$ | 29.78 |
> |  | Ind. | 90,562 | OOM | $\infty$ | 215,220 | OOM | $\infty$ |
> |  | Prod. | 100,625 | OOM | $\infty$ | 239,134 | OOM | $\infty$ |
> | LightHGNN | Trans. | 10,063 | 41.82$_{\pm 3.48}$ | 0.44 | 23,914 | 38.48$_{\pm 5.59}$ | 0.67 |
> |  | Ind. | 90,562 | 41.15$_{\pm 3.23}$ | 2.18 | 215,220 | 37.62$_{\pm 5.89}$ | 5.05 |
> |  | Prod. | 100,625 | 41.22$_{\pm 3.26}$ | 2.4 | 239,134 | 37.70$_{\pm 5.86}$ | 5.62 |
> | LightHGNN+ | Trans. | 10,063 | **42.50**$_{\pm 3.74}$ | 0.44 | 23,914 | **38.76**$_{\pm 5.24}$ | 0.67 |
> |  | Ind. | 90,562 | **42.28**$_{\pm 3.62}$ | 2.18 | 215,220 | **38.26**$_{\pm 5.67}$ | 5.05 |
> |  | Prod. | 100,625 | **42.31**$_{\pm 3.63}$ | 2.4 | 239,134 | **38.31**$_{\pm 5.62}$ | 5.62 |
>
> As shown in the table, our LightHGNNs can not only achieve better performance but also extremely reduce the inference time. As the scale of the dataset increases, the advantage of our LightHGNNs becomes more obvious. Naive HGNN[2] becomes slower and slower, even throwing the Out-Of-Memory error confronting 10w+ vertices, while our LightHGNNs are still fast. Besides, our LightHGNNs exhibit better performance compared to the naive HGNN under the transductive testing set (about 1w or 2w vertices for testing), which demonstrates the effectiveness of our distillation framework. The results indicate that our LightHGNNs are more suitable for large-scale hypergraph inference.
>
> [1] Li et al. SHARE: a System for Hierarchical Assistive Recipe Editing. EMNLP, 2022.
>
> [2] Feng et al. Hypergraph Neural Networks. AAAI, 2019.

---

> ### Author Response · Authors · 2023-11-19
> **Response to Reviewer U1AB (Part 3/3)**
>
> ### **Response to W5**
> We apologize for the unclear descriptions. Compared to the graph KD methods, our LightHGNN can capture high-order information. This is achieved by the proposed high-order reliable quantification, high-order correlation sampling, and high-order soft-target constraint. First, our high-order reliable quantification module can endow each hyperedge with a reliable score by evaluating its capability to combat noise. Second, our high-order correlation sampling module can sample the high-order correlations from the hypergraph via the reliable score, which is implemented by sampling from the Bernoulli distribution. Third, our high-order soft-target constraint can supervise the learned embeddings from the student MLPs and explicitly pull them close to the soft labels of vertices from those reliable hyperedges. All three modules can help the student MLPs capture the high-order information, thus yielding better performance on hypergraph datasets. We will add those descriptions in the revised version.
>
> We also summarize the challenges of hypergraph KD compared to graph KD, which states the necessity of exploring the hypergraph KD methods.
> - **The beyond pairwise modeling capability brings more challenges.** The hypergraph is a generalization of the graph, which can model the low-order structure (graph) and high-order structure. Thus, compared to the distilling graph neural networks that only consider the pair-wise structure, distilling hypergraph neural networks will consider both low-order and high-order structures, which brings more challenges to the distillation of hypergraph neural networks.
> - **The multi-model property of hypergraph brings more challenges.** Hypergraph can naturally model the multi-modal correlation via the incidence matrix concatenation [1], which is more flexible than the graph. In real-world applications, the complex high-order correlations are inherently multi-modal. For example, in the social network, the people and relationships are represented by vertex and hyperedge, and the people can be divided into different groups, and the relationships can be divided into different types, like family, friends, and colleagues. Clearly, distilling those complex (multi-modal) high-order correlations (hypergraph neural networks) is more challenging than the pairwise correlation (graph neural networks).
> - **The information of the hypergraph is hard to quantify.** As for distilling graph neural networks, the information of the graph is easy to quantify. For example, the degree of the edge is fixed to 2, and the connection of the vertex is simple (single modality and directly propagating messages from other vertices). In contrast, the information in the hypergraph is hard to quantify. For example, the degree of the hyperedge is not fixed (one, two, or more are all legal), and the connection of the vertex is complex (multi-modal and propagating messages from other vertices or hyperedge or group vertices). Thus, controlling the process of distilling hypergraph neural networks or making a reliable hypergraph neural network distillation is more challenging than distilling graph neural networks.
>
> As demonstrated in [2], GCN [3] is a special case of  HGNN confronting the pair-wise correlation. HGNN is more powerful than GNN but also suffers from the higher runtime and memory cost. Thus, to better capture those high-order correlations in the hypergraph, we propose a reliable hyperedge quantification and sampling strategy to achieve topology-aware distillation for hypergraph neural networks. This work is the first attempt at distilling hypergraph neural networks. In the future, we will consider more downstream tasks, more modalities, and more strategies of reliable quantification for distilling hypergraph neural networks.
>
> [1] Feng et al. Hypergraph Neural Networks. AAAI, 2019.
>
> [2] Gao et al. HGNN$^+$: General Hypergraph Neural Networks. TPAMI, 2023.
>
> [3] Kipf et al. Semi-supervised classification with graph convolutional networks. ICLR, 2017.

---

> > ### Comment · Reviewer_U1AB · 2023-11-21
> >
> > Thank you for your response. I have carefully reviewed the rebuttal provided, as well as the insights from other reviews. I‘ve decided to raise my score to 6.

---

> > > ### Author Response · Authors · 2023-11-21
> > > **Response to Reviewer U1AB**
> > >
> > > Thanks for your feedback. We are glad to hear that your concerns have been addressed by our explanations. We will add those revisions to the camera-ready version.
> > >
> > > Thanks again for your careful review and valuable comments to help us improve our submission.

---

### Official Review · Reviewer_PX2s · 2023-10-19

**Soundness:** 3 good
**Presentation:** 3 good
**Contribution:** 2 fair
**Rating:** 6
**Confidence:** 5

**Summary:**

In this study, the authors introduce LightHGNN and LightHGNN+, two models aimed at enhancing the efficiency of Hypergraph Neural Networks.

These models bridge the gap with Multi-Layer Perceptrons (MLPs), eliminating hypergraph dependencies and significantly improving computational efficiency.

Experimental results demonstrate that LightHGNNs achieve competitive or superior performance, showcasing their effectiveness and speed in hypergraph-based inference.

**Strengths:**

1. The paper is well-structured and the proposed methods including reliable hyperedge quantification and sampling strategy are clearly explained, with the appendix giving additional relevant details.
2. Clearly labelled figures and visualisations, such as Figures 2, 3, S1, S2, enhance the comprehension of the presented concepts.
3. The efficacy of LightHGNN in knowledge distillation is demonstrated across three settings: transductive, inductive, and production.
4. The proposed methods showcase inventive problem-solving abilities by filling gaps in existing approaches, particularly in the domain of knowledge distillation for hypergraphs.

**Weaknesses:**

1. The paper only evaluates a single baseline model for hypergraphs, Hypergraph Neural Network [Feng et al., 2019], without exploring knowledge distillation in more recent advanced methods that compute hyperedge embeddings [e.g., Wang et al. (2023), Chien et al. (2022)].
2. Additional experiments, particularly investigating deep LightHGNN$^{+}$ models with varying hidden layers (e.g., 2, 3, ..., 10), are necessary to substantiate the assertion that LightHGNN$^{+}$ effectively combats over-smoothing.
3. The datasets utilised in this research, as outlined in Table S2, are relatively small in size, which diminishes the persuasiveness of knowledge distillation due to the absence of large-scale data.


References
* [Feng et al., 2019] Hypergraph Neural Networks, AAAI'19
* [Wang et al., 2023]: Equivariant Hypergraph Diffusion Neural Operators, ICLR'23
* [Chien et al., 2022]: You are AllSet: A Multiset Function Framework for Hypergraph Neural Networks, ICLR'22

**Questions:**

1. What criteria were considered while selecting the baseline Hypergraph Neural Network?
2. How does this restricted choice impact the overall diversity and representation of neural models applied to hypergraphs?
3. Were there any specific metrics or criteria that would be considered as indicators of effective combating of over-smoothing in the context of deep LightHGNN$^{+}$ models with varying hidden layers, e.g., 2, 3, ..., 10?
4. In the event that the experiments confirm the effectiveness of deep LightHGNN$^{+}$ models in combating over-smoothing, what implications might this have for the broader research community?
5. How might findings on over-smoothing resistance inform future research directions or practical applications in the context of knowledge distillation from hypergraph neural networks?
6. Given the small size of the datasets, what steps were taken to ensure that the findings and conclusions drawn from these datasets can be generalised to larger, real-world scenarios, e.g., (hyper)graphs that cannot fit into GPU memory?

---

> ### Author Response · Authors · 2023-11-19
> **Response to Reviewer PX2s (Part 1/4)**
>
> ### **Response to W1**
> Thanks for your comments. In the submission, we have provided the experiments on different teacher architectures, including HGNN, HGNN+, UniGNN, UniGAT, in Appendix H and Table S9. Here, following the reviewer's suggestions, we further conduct experiments with more advanced methods including UniGCN2[1], ED-HNN[2], and AllSet[3]. Experimental results are shown in the following table.
>
> | Method | Setting | Original | LightHGNN | LightHGNN+ |
> |:---:|:---:|:---:|:---:|:---:|
> | HGNN | Prod. | 71.52$_{\pm 1.31}$ | 71.14$_{\pm 1.23}$ | **71.69**$_{\pm 1.44}$ |
> |  | Tran. | 70.75$_{\pm 1.49}$ | 70.88$_{\pm 1.29}$ | **71.40**$_{\pm 1.50}$ |
> |  | Ind. | 72.72$_{\pm 2.32}$ | 72.22$_{\pm 2.08}$ | **72.86**$_{\pm 2.33}$ |
> | HGNN+ | Prod. | **72.91**$_{\pm 1.23}$ | 72.63$_{\pm 0.80}$ | 72.65$_{\pm 0.66}$ |
> |  | Tran. | 71.83$_{\pm 1.50}$ | 72.33$_{\pm 0.89}$ | **72.47**$_{\pm 0.95}$ |
> |  | Ind. | **74.43**$_{\pm 2.39}$ | 73.84$_{\pm 1.69}$ | 74.15$_{\pm 1.66}$ |
> | UniGNN | Prod. | 72.73$_{\pm 1.45}$ | 72.66$_{\pm 1.48}$ | **72.83**$_{\pm 1.66}$ |
> |  | Tran. | 71.88$_{\pm 1.51}$ | 72.38$_{\pm 1.54}$ | **72.40**$_{\pm 1.66}$ |
> |  | Ind. | 73.79$_{\pm 2.57}$ | 73.79$_{\pm 1.68}$ | **74.54**$_{\pm 2.58}$ |
> | UniGAT | Prod. | **73.00**$_{\pm 1.40}$ | 72.74$_{\pm 1.20}$ | 72.59$_{\pm 1.32}$ |
> |  | Tran. | 71.99$_{\pm 1.49}$ | **72.46**$_{\pm 1.36}$ | 72.32$_{\pm 1.21}$ |
> |  | Ind. | **74.29**$_{\pm 2.53}$ | 73.90$_{\pm 1.69}$ | 73.65$_{\pm 2.64}$ |
> | UniGCN2[1] | Prod. | 72.94$_{\pm 0.97}$ | 73.29$_{\pm 1.29}$ | **73.58**$_{\pm 1.69}$ |
> |  | Tran. | 72.04$_{\pm 0.84}$ | 72.87$_{\pm 1.12}$ | **73.05**$_{\pm 1.60}$ |
> |  | Ind. | 73.81$_{\pm 2.48}$ | 74.95$_{\pm 2.30}$ | **75.71**$_{\pm 2.54}$ |
> | ED-HNN[2] | Prod. | 72.01$_{\pm 1.06}$ | 73.45$_{\pm 1.48}$ | **73.53**$_{\pm 1.61}$ |
> |  | Tran. | 70.64$_{\pm 1.52}$ | 72.11$_{\pm 1.66}$ | **73.34**$_{\pm 1.78}$ |
> |  | Ind. | 73.79$_{\pm 1.72}$ | **74.79**$_{\pm 1.61}$ | 74.32$_{\pm 1.86}$ |
> | AllSet[3] | Prod. | 71.37$_{\pm 1.60}$ | 73.20$_{\pm 1.96}$ | **73.53**$_{\pm 1.48}$ |
> |  | Tran. | 71.15$_{\pm 1.93}$ | 73.00$_{\pm 2.57}$ | **73.14**$_{\pm 1.32}$ |
> |  | Ind. | 72.97$_{\pm 2.61}$ | **75.36**$_{\pm 2.07}$ | 75.10$_{\pm 2.61}$ |
>
> As shown in the table, as the teacher model becomes more advanced, the performance of our LightHGNNs also becomes better. We will add those results in Table S9 "Experimental results of LightHGNNs with different teacher architectures." in Appendix H.
>
> [1] Huang et al. Unignn: a unified framework for graph and hypergraph neural networks. IJCAI, 2021.
>
> [2] Wang et al. Equivariant Hypergraph Diffusion Neural Operators, ICLR, 2023.
>
> [3] Chien et al. You are AllSet: A Multiset Function Framework for Hypergraph Neural Networks, ICLR, 2022.
>
>
> ### **Response to W2**
> Thanks for your suggestions. We further conduct experiments on the number of hidden layers of MLP, HGNN and LightHGNNs. The results are shown in the following table.
>
> | #Layer | MLP | HGNN | LightHGNN | LightHGNN+ |
> |---|---|---|---|---|
> | 2 | 40.87$_{\pm 1.43}$ | 50.78$_{\pm 1.67}$ | 50.19$_{\pm 1.56}$ | 50.47$_{\pm  1.92}$ |
> | 3 | 41.25$_{\pm 1.03}$ | 50.32$_{\pm 1.23}$ | 49.52$_{\pm 2.38}$ | 50.38$_{\pm  2.23}$ |
> | 4 | 41.44$_{\pm 1.45}$ | 44.97$_{\pm 3.72}$ | 47.58$_{\pm 5.82}$ | 49.75$_{\pm  2.91}$ |
> | 5 | 41.52$_{\pm 0.67}$ | 42.59$_{\pm 2.45}$ | 48.55$_{\pm 5.59}$ | 47.97$_{\pm 6.08}$ |
> | 6 | 41.18$_{\pm 0.93}$ | 39.72$_{\pm 3.76}$ | 47.61$_{\pm 5.95}$ | 47.66$_{\pm 5.84}$ |
> | 7 | 40.23$_{\pm 1.48}$ | 31.47$_{\pm 5.61}$ | 46.33$_{\pm 8.06}$ | 47.67$_{\pm 5.97}$ |
> | 8 | 38.75$_{\pm 1.95}$ | 32.89$_{\pm 4.11}$ | 46.64$_{\pm 7.89}$ | 47.16$_{\pm 6.31}$ |
> | 9 | 36.81$_{\pm 2.22}$ | 32.14$_{\pm 4.59}$ | 46.24$_{\pm 5.50}$ | 47.69$_{\pm 6.03}$ |
> | 10 | 36.75$_{\pm 2.35}$ | 32.91$_{\pm 0.89}$ | 42.64$_{\pm 6.11}$ | 44.18$_{\pm 5.99}$ |
>
> As shown in the table, the accuracy of MLP, HGNN, and LightHGNNs all significantly decrease as the number of layers increases, especially for the MLP and HGNN. However, the accuracy of LightHGNN+ is more stable than MLP, HGNN, and LightHGNNs. This is because the proposed LightHGNN+ can alleviate the over-smoothing phenomenon via the hyperedge reliability sampling, thus yielding robust performance combating the increase of the number of layers.

---

> ### Author Response · Authors · 2023-11-19
> **Response to Reviewer PX2s (Part 2/4)**
>
> ### **Response to W3**
> Thanks for your suggestions. We further conduct experiments on two large-scale hypergraph datasets[1]: Recipe-100k (10w vertices) and Recipe-200k (24w vertices). The two datasets both contain eight categories. Specifically, the Recipe-100k dataset contains 101,585 vertices and 12,387 hyperedges, and the Recipe-200k dataset contains 240,094 vertices and 18,129 hyperedges. The vertex features are the bag of words from the sentence that makes the recipe. Hyperedges are the ingredients of the recipe or the Keywords for food preparation steps. Experiments on the two datasets are shown in the following table.
> |  |  | Recipe-100k |  |  | Recipe-200k |  |  |
> |---|---|:---:|:---:|:---:|:---:|:---:|:---:|
> |  |  | #Vertices of Testing | Acc.(%) | Infer Time(s) | #Vertices of Testing | Acc.(%) | Infer Time(s) |
> | HGNN | Trans. | 10,063 | 41.26$_{\pm 3.08}$ | 5.39 | 23,914 |  37.25$_{\pm 4.88}$ | 29.78 |
> |  | Ind. | 90,562 | OOM | $\infty$ | 215,220 | OOM | $\infty$ |
> |  | Prod. | 100,625 | OOM | $\infty$ | 239,134 | OOM | $\infty$ |
> | LightHGNN | Trans. | 10,063 | 41.82$_{\pm 3.48}$ | 0.44 | 23,914 | 38.48$_{\pm 5.59}$ | 0.67 |
> |  | Ind. | 90,562 | 41.15$_{\pm 3.23}$ | 2.18 | 215,220 | 37.62$_{\pm 5.89}$ | 5.05 |
> |  | Prod. | 100,625 | 41.22$_{\pm 3.26}$ | 2.4 | 239,134 | 37.70$_{\pm 5.86}$ | 5.62 |
> | LightHGNN+ | Trans. | 10,063 | **42.50**$_{\pm 3.74}$ | 0.44 | 23,914 | **38.76**$_{\pm 5.24}$ | 0.67 |
> |  | Ind. | 90,562 | **42.28**$_{\pm 3.62}$ | 2.18 | 215,220 | **38.26**$_{\pm 5.67}$ | 5.05 |
> |  | Prod. | 100,625 | **42.31**$_{\pm 3.63}$ | 2.4 | 239,134 | **38.31**$_{\pm 5.62}$ | 5.62 |
>
> As shown in the table, our LightHGNNs can not only achieve better performance but also extremely reduce the inference time. As the scale of the dataset increases, the advantage of our LightHGNNs becomes more obvious. Naive HGNN[2] becomes slower and slower even throwing Out-Of-Memory error confronting 10w+ vertices, while our LightHGNNs are still fast. Besides, our LightHGNNs exhibit better performance compared to the naive HGNN under the transductive testing set (about 1w or 2w vertices for testing), which demonstrates the effectiveness of our distillation framework. The results indicate that our LightHGNNs are more suitable for large-scale hypergraph inference.
>
> [1] Li et al. SHARE: a System for Hierarchical Assistive Recipe Editing. EMNLP, 2022.
>
> [2] Feng et al. Hypergraph Neural Networks. AAAI, 2019.
>
>
> ### **Response to Q1**
> Thanks for your comments. The main reason for choosing the baseline HGNN is that it is a typical and widely used neural network on hypergraphs. It is also a direct extension of the typical graph neural networks [1] as demonstrated in Gao et al. [2]. Since the proposed LightHGNNs is a general distillation framework for hypergraph neural networks, we also provide experiments on other advanced hypergraph neural networks, including HGNN+[2], UniGNN, UniGAT, UniGCN2[3], ED-HNN[4], and AllSet[5], as shown in the table of "Response to W1".
>
> [1] Kipf et al. Semi-supervised classification with graph convolutional networks. ICLR, 2017.
>
> [2] Gao et al. HGNN$^+$: General Hypergraph Neural Networks. TPAMI, 2023.
>
> [3] Huang et al. Unignn: a unified framework for graph and hypergraph neural networks. IJCAI, 2021.
>
> [4] Wang et al. Equivariant Hypergraph Diffusion Neural Operators, ICLR, 2023.
>
> [5] Chien et al. You are AllSet: A Multiset Function Framework for Hypergraph Neural Networks, ICLR, 2022.
>
> ### **Response to Q2**
> Thanks for your comments. In this paper, we propose a hypergraph neural networks distillation framework, which does not rely on the specific hypergraph neural networks selection. The proposed LightHGNN can be applied to any hypergraph neural network to achieve fast and accurate inference, as shown in the table of "Response to W1".

---

> ### Author Response · Authors · 2023-11-19
> **Response to Reviewer PX2s (Part 3/4)**
>
> ### **Response to Q3**
> Thanks for your comments. Actually, in our experiment section 5.4, we attempt to define a specific metric to quantify the over-smoothing phenomenon. Here, we design a topology-aware score $\mathcal{S}$ to measure the relevance of features and hypergraph typology. Specifically, we first calculate the cosine similarity between the vertex features and the hyperedge features, and then calculate the average cosine similarity of all vertices in the hypergraph. The topology-aware score $\mathcal{S}$ is defined as the average cosine similarity of all hyperedges in the hypergraph. The higher the score, the more relevant the features and hypergraph typology. We first provide the accuracy of MLP, HGNN and LightHGNNs with respect to different number of layers, as shown in the following table.
>
> | #Layer | MLP | HGNN | LightHGNN | LightHGNN+ |
> |---|---|---|---|---|
> | 2 | 40.87$_{\pm 1.43}$ | 50.78$_{\pm 1.67}$ | 50.19$_{\pm 1.56}$ | 50.47$_{\pm  1.92}$ |
> | 3 | 41.25$_{\pm 1.03}$ | 50.32$_{\pm 1.23}$ | 49.52$_{\pm 2.38}$ | 50.38$_{\pm  2.23}$ |
> | 4 | 41.44$_{\pm 1.45}$ | 44.97$_{\pm 3.72}$ | 47.58$_{\pm 5.82}$ | 49.75$_{\pm  2.91}$ |
> | 5 | 41.52$_{\pm 0.67}$ | 42.59$_{\pm 2.45}$ | 48.55$_{\pm 5.59}$ | 47.97$_{\pm 6.08}$ |
> | 6 | 41.18$_{\pm 0.93}$ | 39.72$_{\pm 3.76}$ | 47.61$_{\pm 5.95}$ | 47.66$_{\pm 5.84}$ |
> | 7 | 40.23$_{\pm 1.48}$ | 31.47$_{\pm 5.61}$ | 46.33$_{\pm 8.06}$ | 47.67$_{\pm 5.97}$ |
> | 8 | 38.75$_{\pm 1.95}$ | 32.89$_{\pm 4.11}$ | 46.64$_{\pm 7.89}$ | 47.16$_{\pm 6.31}$ |
> | 9 | 36.81$_{\pm 2.22}$ | 32.14$_{\pm 4.59}$ | 46.24$_{\pm 5.50}$ | 47.69$_{\pm 6.03}$ |
> | 10 | 36.75$_{\pm 2.35}$ | 32.91$_{\pm 0.89}$ | 42.64$_{\pm 6.11}$ | 44.18$_{\pm 5.99}$ |
>
> As shown in the table, the accuracy of MLP, HGNN, and LightHGNNs all significantly decrease as the number of layers increases, especially for the MLP and HGNN. However, the accuracy of LightHGNN+ is more stable than MLP, HGNN, and LightHGNNs. This is because the proposed LightHGNN+ can alleviate the over-smoothing phenomenon via the hyperedge reliability sampling, thus yielding robust performance combating the increase of the number of layers. We further provide the topology-aware score $\mathcal{S}$ of MLP, HGNN and LightHGNNs with respect to different number of layers, as shown in the following table.
>
> | #Layer | MLP | HGNN | LightHGNN | LightHGNN+ |
> |---|---|---|---|---|
> | 2 | 1.180 | 0.040 | 0.713 | 0.770 |
> | 3 | 2.498 | 0.049 | 0.876 | 0.954 |
> | 4 | 2.660 | 0.017 | 0.782 | 0.821 |
> | 5 | 2.778 | 0.023 | 0.533 | 0.648 |
> | 6 | 2.887 | 0.021 | 0.454 | 0.570 |
> | 7 | 1.922 | 0.008 | 0.076 | 0.376 |
> | 8 | 2.842 | 0.004 | 0.082 | 0.226 |
> | 9 | 2.480 | 0.007 | 0.058 | 0.178 |
> | 10 | 2.096 | 0.006 | 0.014 | 0.120 |
>
> As shown in the above table, we have the following three observations. First, despite the increase in the number of layers, the topology-aware score of MLP is always higher than that of HGNN and LightHGNNs. This is because the MLP only relies on the vertex features and ignores the hypergraph typology. Second, the topology-aware score of LightHGNN+ is more stable than MLP, HGNN, and LightHGNNs. This is because the proposed LightHGNN+ can alleviate the over-smoothing phenomenon via the hyperedge reliability sampling, thus yielding robust performance combating the increase of the number of layers. Third, we find an interesting phenomenon that the topology-aware score significantly decreases in #layer 6->7. The results explain when the accuracy of HGNN and LightHGNN significantly decreases in #layer 6->7, as shown in the above table. This is because the phenomenon of over-smoothing has reached a critical point with the increase in the number of layers, impacting the representations for hypergraphs. As a result, the performance has sharply declined. However, why the critical point is reached at this specific layer needs further investigation in future work. I believe this is an intriguing experimental result that will impact the research community's study of over-smoothing.
>
> ### **Response to Q4**
> Thanks for your insightful comments. It's important to note that the over-smoothing metric we introduced is a preliminary attempt to capture the phenomenon. The defined metrics attempt to measure the relevance of features and hypergraph typology. However, the over-smoothing phenomenon is a complex problem, which can be caused by various reasons like the dataset, model architecture, and optimization strategy. Our typology-aware scoring is a preliminary attempt, and there is much work to be done further, such as considering the weights between different nodes within the same hyperedge and considering the influence of the high-order neighbor smoothing. These avenues provide various opportunities for researchers to delve deeper into the intricacies of over-smoothing and its implications for representation learning on hypergraphs.

---

> ### Author Response · Authors · 2023-11-19
> **Response to Reviewer PX2s (Part 4/4)**
>
> ### **Response to Q5**
> Thanks for your insightful comments. In the field of knowledge distillation for Hypergraph Neural Networks (HGNNs), the findings regarding resistance to over-smoothing could impact future research directions and practical applications in the following aspects:
>
> 1. **Model Robustness**: Research on resistance mechanisms to over-smoothing can aid in developing more robust HGNN models that maintain feature diversity across multiple network layers and avoid homogenization of features in the deeper layers.
>
> 2. **Deep Network Design**: Understanding how to control and mitigate over-smoothing is crucial for designing deeper HGNN structures while preserving high-order feature information, enhancing the expressiveness of the models.
>
> 3. **Computational Efficiency**: Over-smoothing resistance techniques could reduce the number of iterations needed during distillation, as they help avoid rapid convergence to suboptimal states. This directly impacts the computational efficiency and inference speed of models, especially on large-scale datasets.
>
> 4. **Optimization Strategy Research**: Research into over-smoothing could inspire new optimization strategies, such as regularization techniques or novel loss functions, which could reduce the issue of over-smoothing while maintaining model performance.
>
> These implications suggest that research into over-smoothing resistance is not only vital for understanding HGNNs but also for enhancing the practical deployment value of the distilled student models.
>
> ### **Response to Q6**
> Thanks for your suggestions. We further conduct experiments to demonstrate the capability of our LightHGNNs on large-scale hypergraph inference. Two large scale hypergraph datasets[1]: Recipe-100k (10w vertices) and Recipe-200k (24w vertices) are employed here. The two datasets both contain eight categories. Specifically, the Recipe-100k dataset contains 101,585 vertices and 12,387 hyperedges, and the Recipe-200k dataset contains 240,094 vertices and 18,129 hyperedges. The vertex features are the bag of words from the sentence that makes the recipe. Hyperedges are the ingredients of the recipe or the Keywords for food preparation steps. Experiments on the two datasets are shown in the following table.
>
> |  |  | Recipe-100k |  |  | Recipe-200k |  |  |
> |---|---|:---:|:---:|:---:|:---:|:---:|:---:|
> |  |  | #Vertices of Testing | Acc.(%) | Infer Time(s) | #Vertices of Testing | Acc.(%) | Infer Time(s) |
> | HGNN | Trans. | 10,063 | 41.26$_{\pm 3.08}$ | 5.39 | 23,914 |  37.25$_{\pm 4.88}$ | 29.78 |
> |  | Ind. | 90,562 | OOM | $\infty$ | 215,220 | OOM | $\infty$ |
> |  | Prod. | 100,625 | OOM | $\infty$ | 239,134 | OOM | $\infty$ |
> | LightHGNN | Trans. | 10,063 | 41.82$_{\pm 3.48}$ | 0.44 | 23,914 | 38.48$_{\pm 5.59}$ | 0.67 |
> |  | Ind. | 90,562 | 41.15$_{\pm 3.23}$ | 2.18 | 215,220 | 37.62$_{\pm 5.89}$ | 5.05 |
> |  | Prod. | 100,625 | 41.22$_{\pm 3.26}$ | 2.4 | 239,134 | 37.70$_{\pm 5.86}$ | 5.62 |
> | LightHGNN+ | Trans. | 10,063 | **42.50**$_{\pm 3.74}$ | 0.44 | 23,914 | **38.76**$_{\pm 5.24}$ | 0.67 |
> |  | Ind. | 90,562 | **42.28**$_{\pm 3.62}$ | 2.18 | 215,220 | **38.26**$_{\pm 5.67}$ | 5.05 |
> |  | Prod. | 100,625 | **42.31**$_{\pm 3.63}$ | 2.4 | 239,134 | **38.31**$_{\pm 5.62}$ | 5.62 |
>
> As shown in the table, our LightHGNNs can not only achieve better performance but also extremely reduce the inference time. As the scale of the dataset increases, the advantage of our LightHGNNs becomes more obvious. Naive HGNN[2] becomes slower and slower, even throwing the Out-Of-Memory error confronting 10w+ vertices, while our LightHGNNs are still fast. Besides, our LightHGNNs exhibit better performance compared to the naive HGNN under the transductive testing set (about 1w or 2w vertices for testing), which demonstrates the effectiveness of our distillation framework. The results indicate that our LightHGNNs are more suitable for large-scale hypergraph inference.
>
> [1] Li et al. SHARE: a System for Hierarchical Assistive Recipe Editing. EMNLP, 2022.
> [2] Feng et al. Hypergraph Neural Networks. AAAI, 2019.

---

> > ### Comment · Reviewer_PX2s · 2023-11-23
> > **Thanks for the responses**
> >
> > Thanks for the additional experiments. After carefully examining all the reviews and their responses, my level of confidence regarding the evaluation has risen. Please add the additional experiments and provide a summary of the main insights and high-level messages in a subsequent version of the paper.

---

### Official Review · Reviewer_uHuc · 2023-10-30

**Soundness:** 3 good
**Presentation:** 3 good
**Contribution:** 3 good
**Rating:** 6
**Confidence:** 2

**Summary:**

The paper proposes a method called LightHGNN to enhance the efficiency of Hypergraph Neural Networks (HGNNs). The proposed LightHGNN bridges the gap between HGNNs and Multi-Layer Perceptrons (MLPs) to eliminate the dependency on hypergraph structure during inference, reducing computational complexity and improving inference speed. LightHGNN distills knowledge from teacher HGNNs to student MLPs using soft labels. Additionally, LightHGNN+ injects reliable high-order correlations into the student MLPs to achieve topology-aware distillation and resistance to over-smoothing. Experimental results show that LightHGNNs achieve competitive or better performance than HGNNs, even without hypergraph dependency.

**Strengths:**

The idea of using hyperedge quantification and sampling is interesting.

The paper is clear and easy to follow.

**Weaknesses:**

The idea of using distillation to improve the efficiency of neural networks is not new, even in the field of Graph Neural Networks. This work is an implementation of this idea to Hypergraph Neural Networks. Though the author claims that some special designs should be considered as the technical contributions of this paper, the general novelty of this paper is borderline.

**Questions:**

1. Though the  LightHGNN and  LightHGNN+ models are distilled from the HGNN model, their performance is sometimes even better than the original teacher HGNN. May the author explain why this happens?

2. The author claims that LightHGNN+ is able to capture the topology information, it is expected that  LightHGNN+ should thus perform better than  LightHGNN. However, there is not general superiority of  LightHGNN+ over  LightHGNN. May the author explain why?

---

> ### Author Response · Authors · 2023-11-19
> **Response to Reviewer uHuc (Part 1/2)**
>
> ### **Response to Weaknesses**
> Thanks for your comments. We apologize for the unclear description of the novelty of this paper. In the following, we clarify the novelty of the proposed LightHGNNs from two perspectives.
> - **Compared to GNNs-to-MLPs Methods.** First, the Graph-Less Neural Networks (GLNN) [1] proposes using the soft label of the teacher GNNs to supervise the student MLPs. Compared to it, our LightHGNN is a simple extension from GNNs to HGNNs. However, our LightHGNN$^+$ further proposes the high-order soft label to help the student MLPs learn more high-order structure information from the teacher HGNNs. Second, as for the Knowledge-inspired Reliable Distillation (KRD) [2], its student MLPs can only be supervised by the soft label from those reliable vertices (knowledge point in their paper), which still lose the structure information and cannot be utilized in the hypergraph. However, our LightHGNN$^+$ can quantify the reliability of the high-order correlations and inject the reliable high-order information into the student MLPs via explicit supervision from high-order soft labels of those reliable hyperedges. The proposed LightHGNN$^+$ can sufficiently take advantage of both the vertex features and high-order structure information, thus yielding better performance and faster inference speed.
> - **Compared to HGNNs Methods.** To the best of my belief, our work is the first to distill hypergraph neural networks to MLPs. Our LightHGNNs can achieve comparable and even better performance in hypergraph datasets with much faster inference speed and lower memory cost. Those capabilities allow the LightHGNNs to be applied to large-scale hypergraph inference, which can not be achieved by the naive HGNNs. In the following table, we conduct experiments on two larger hypergraph datasets[3]: Recipe-100k (10w vertices) and Recipe-200k (24w vertices).
>
> |  |  | Recipe-100k |  |  | Recipe-200k |  |  |
> |---|---|:---:|:---:|:---:|:---:|:---:|:---:|
> |  |  | #Vertices of Testing | Acc.(%) | Infer Time(s) | #Vertices of Testing | Acc.(%) | Infer Time(s) |
> | HGNN | Trans. | 10,063 | 41.26$_{\pm 3.08}$ | 5.39 | 23,914 |  37.25$_{\pm 4.88}$ | 29.78 |
> |  | Ind. | 90,562 | OOM | $\infty$ | 215,220 | OOM | $\infty$ |
> |  | Prod. | 100,625 | OOM | $\infty$ | 239,134 | OOM | $\infty$ |
> | LightHGNN | Trans. | 10,063 | 41.82$_{\pm 3.48}$ | 0.44 | 23,914 | 38.48$_{\pm 5.59}$ | 0.67 |
> |  | Ind. | 90,562 | 41.15$_{\pm 3.23}$ | 2.18 | 215,220 | 37.62$_{\pm 5.89}$ | 5.05 |
> |  | Prod. | 100,625 | 41.22$_{\pm 3.26}$ | 2.4 | 239,134 | 37.70$_{\pm 5.86}$ | 5.62 |
> | LightHGNN+ | Trans. | 10,063 | **42.50**$_{\pm 3.74}$ | 0.44 | 23,914 | **38.76**$_{\pm 5.24}$ | 0.67 |
> |  | Ind. | 90,562 | **42.28**$_{\pm 3.62}$ | 2.18 | 215,220 | **38.26**$_{\pm 5.67}$ | 5.05 |
> |  | Prod. | 100,625 | **42.31**$_{\pm 3.63}$ | 2.4 | 239,134 | **38.31**$_{\pm 5.62}$ | 5.62 |
>
> As shown in the table, our LightHGNNs can not only achieve better performance but also extremely reduce the inference time. As the scale of the dataset increases, the advantage of our LightHGNNs becomes more obvious. Naive HGNN[4] becomes slower and slower even throwing Out-Of-Memory error confronting 10w+ vertices, while our LightHGNNs are still fast. Besides, our LightHGNNs exhibit better performance compared to the naive HGNN under the transductive testing set (about 1w or 2w vertices for testing), which demonstrates the effectiveness of our distillation framework.
>
> [1] Zhang et al. Graph-less neural networks: Teaching old MLPs new tricks via distillation. ICLR, 2022.
>
> [2] Wu et al. Quantifying the knowledge in gnns for reliable distillation into MLPs. ICML, 2023.
>
> [3] Li et al. SHARE: a System for Hierarchical Assistive Recipe Editing. EMNLP, 2022.
>
> [4] Feng et al. Hypergraph Neural Networks. AAAI, 2019.

---

> ### Author Response · Authors · 2023-11-19
> **Response to Reviewer uHuc (Part 2/2)**
>
> ### **Response to Q1**
> Thanks for your comments. We attribute the performance improvement of LightHGNN$^+$ to the following two reasons. First, knowledge distillation from teacher HGNN determines the student LightHGNNs can achieve comparable performance to HGNN. Second, the proposed hyperedge reliability sampling can help the student MLPs distinguish those noise hyperedges in the original hypergraph, which can help the student LightHGNN$^+$ capture more task-related and reliable information, alleviate over-smoothing (as empirically demonstrated in Section 5.4), thus yielding better performance.
>
> ### **Response to Q2**
> Thanks for your comments. Actually, in most datasets under the transductive setting, LightHGNN$^+$ performs better than LightHGNN. There are still three datasets (News20, CA-Cora, and DBLP-Conf), where LightHGNN$^+$ performs worse than LightHGNN and HGNN. This is because, as shown in the following table, the number of hyperedges in the three datasets is small, which limits the power of our reliability hyperedge quantification and sampling module. However, as the number of hyperedges increases, the performance of LightHGNN$^+$ becomes better and better. As shown in the above table, our LightHGNN$^+$ can achieve better performance than LightHGNN and HGNN in the Recipe-100k and Recipe-200k datasets. Thus, our LightHGNN$^+$ has more potential to be applied to large-scale hypergraph inference and achieve better performance.
>
> |  | #Vertices | #Edge/Hyperedge | #Features |
> |---|---|---|---|
> | **DBLP-Conf** | 4057 | **20** | 334 |
> | **News20** | 16342 | **100** | 100 |
> | **CA-Cora** | 2708 | **970** | 1433 |
> | CC-Cora | 2708 | 1483 | 1433 |
> | CC-Citeseer | 3312 | 1004 | 3703 |
> | DBLP-Paper | 4057 | 5701 | 334 |
> | DBLP-Term | 4057 | 6089 | 334 |
> | IMDB-AW | 4278 | 5257 | 3066 |

---

### Official Review · Reviewer_tcvF · 2023-11-02

**Soundness:** 3 good
**Presentation:** 4 excellent
**Contribution:** 3 good
**Rating:** 6
**Confidence:** 4

**Summary:**

The authors demonstrate that a Hypergraph Neural Network (HGNN), which is specifically designed for hypergraph-structured data, can be effectively distilled into a Multi-Layer Perceptron (MLP). To this end, the authors extend a Graph-Neural-Network (GNN) distillation technique (Wu et al. 2023) to accommodate hypergraphs.

**Strengths:**

S1. The paper is well-structured and easy to follow.

S2. It seems that the authors are the first to distill hypergraph neural networks into MLPs, resulting in a significant improvement in speed with only little sacrifice in accuracy.

S3. The proposed method is a logical extension of (Wu et al. 2023)

**Weaknesses:**

W1. First and most importantly, (a) the importance of reliability-based sampling and (b) the effectiveness of the proposed methodology of measuring the reliability need to be demonstrated empirically and/or theoretically. To achieve this, a comparison should be made between the proposed method and alternative approaches, including (a) utilizing all hyperedges without reliability-based sampling and (b) relying on node reliability (Wu et al. 2023).

W2. The empirical results are limited to HGNN, which is one of the most basic hypergraph neural networks. The authors need to investigate the effectiveness of the proposed distillation method across a broader range of hypergraph neural networks , including more advanced ones (e.g., UNIGCN2 and AllSet).

W3. Furthermore, there is scope to explore the generalizability of the proposed method across a wider range of scenarios by incorporating additional downstream tasks (e.g., hyperedge prediction) and diverse datasets. Currently, the most hypergraphs are obtained from bibliographic data.

**Questions:**

Q1. Please address W1.

Q2. Pease address W2.

Q3. Please address W3.

Q4. How did you apply the GNN-distillation methods to hypergraph-structured datasets in the experiments? Please provide details.

Q5. Please elaborate on the distinctive challenges in distilling hypergraph neural networks compared to distilling graph neural networks.

Q6. Please elaborate on the novelty of your approach and its significance when compared to (Wu et al. 2023).

---

> ### Author Response · Authors · 2023-11-19
> **Response to Reviewer tcvF (Part 1/4)**
>
> ### **Response to  W1 and Q1**
> Thanks for your comments. In the following, we use **(a) and (b) to denote** the question *"the importance of reliability-based sampling"* and the question *"the effectiveness of the proposed methodology of measuring the reliability"*, respectively.
>
> **From theoretical point of view.**
> The sampling process of (a) is a typical Bernoulli distribution, whose effectiveness relies on the estimation of hyperedge's reliability (b). In our work, each hyperedge's reliability is generated by the aggregation of linked vertices' variance of entropy. Clearly, the variance of entropy is a good indicator of the reliability of a vertex, which reflects the capability of resisting to noise, as proved by Wu et al.(2023). Thus, in this paper, we extend the concept of vertex's reliability to hyperedge's reliability for high-order correlation reliabile quantification.
>
> **From experimental point of view.**
> Follow the review's suggestion, we further conduct ablation study on "utilizing all hyperedges (a)" and "relying on vertex reliability (b)", as shown in the following table.
>
> |  | HGNN | LightHGNN | LightHGNN+ | (a): LightHGNN (w.o All Hyperedges) | (b): LightHGNN (w.o. Vertex Reliability) |
> |---|---|---|---|---|---|
> | DBLP-Paper | 72.27$_{\pm 0.91}$ | 72.74$_{\pm 1.07}$ | **72.93**$_{\pm 0.97}$ | 72.43$_{\pm 0.88}$ | 72.61$_{\pm 1.07}$ |
> | Coauthor-Cora | 72.82$_{\pm 1.70}$ | 73.63$_{\pm 5.14}$ | **73.54**$_{\pm 3.80}$ | 73.03$_{\pm 3.09}$ | 73.14$_{\pm 3.68}$ |
> | IMDB-AW | 50.47$_{\pm 1.66}$ | 50.19$_{\pm 1.56}$ | **50.78**$_{\pm 1.92}$ | 50.01$_{\pm1.99}$ | 50.42$_{\pm 1.68}$ |
>
> Here, we adopt three hypergraph datasets for experimental validation. The last two columns denote the point (a) and (b), respectively. We have the following two observations. First, we find that the LightHGNN$^+$ (hyperedge reliability) and (b) (vertex reliability) both perform better than LightHGNN without any reliability quantification, which indicates that the reliability quantification of vertex and hyperedge is useful for distilling knowledge from teacher to student. Second, the results of (a) perform worse than both LightHGNN and LightHGNN$^+$. This is because utilizing all hyperedges for high-order knowledge injection will lead to noise injection and over-smoothing problems. Since those hyperedges already learned by the teacher HGNN, we here only need to inject the knowledge of those reliable and task-related hyperedges into the student MLPs. Thus, the proposed hyperedge reliability quantification is crucial for the effectiveness of our distillation framework. This is also the reason that our LightHGNN$^+$ can achieve better performance than LightHGNN.

---

> ### Author Response · Authors · 2023-11-19
> **Response to Reviewer tcvF (Part 2/4)**
>
> ### **Response to  W2 and Q2**
> Thanks for your comments. In this work, we propose a fast-inference framework (including LightHGNN and LightHGNN+) for hypergraph neural networks, whose teachers can be arbitrary hypergraph neural networks. In the following, we provide more experiments of "advanced ones" to show the effectiveness of our distillation framework. As shown in the following table, as the teacher model becomes more advanced, the performance of our LightHGNNs also becomes better.
>
> | Method | Setting | Original | LightHGNN | LightHGNN+ |
> |:---:|:---:|:---:|:---:|:---:|
> | HGNN | Prod. | 71.52$_{\pm 1.31}$ | 71.14$_{\pm 1.23}$ | **71.69**$_{\pm 1.44}$ |
> |  | Tran. | 70.75$_{\pm 1.49}$ | 70.88$_{\pm 1.29}$ | **71.40**$_{\pm 1.50}$ |
> |  | Ind. | 72.72$_{\pm 2.32}$ | 72.22$_{\pm 2.08}$ | **72.86**$_{\pm 2.33}$ |
> | HGNN+ | Prod. | **72.91**$_{\pm 1.23}$ | 72.63$_{\pm 0.80}$ | 72.65$_{\pm 0.66}$ |
> |  | Tran. | 71.83$_{\pm 1.50}$ | 72.33$_{\pm 0.89}$ | **72.47**$_{\pm 0.95}$ |
> |  | Ind. | **74.43**$_{\pm 2.39}$ | 73.84$_{\pm 1.69}$ | 74.15$_{\pm 1.66}$ |
> | UniGNN | Prod. | 72.73$_{\pm 1.45}$ | 72.66$_{\pm 1.48}$ | **72.83**$_{\pm 1.66}$ |
> |  | Tran. | 71.88$_{\pm 1.51}$ | 72.38$_{\pm 1.54}$ | **72.40**$_{\pm 1.66}$ |
> |  | Ind. | 73.79$_{\pm 2.57}$ | 73.79$_{\pm 1.68}$ | **74.54**$_{\pm 2.58}$ |
> | UniGAT | Prod. | **73.00**$_{\pm 1.40}$ | 72.74$_{\pm 1.20}$ | 72.59$_{\pm 1.32}$ |
> |  | Tran. | 71.99$_{\pm 1.49}$ | **72.46**$_{\pm 1.36}$ | 72.32$_{\pm 1.21}$ |
> |  | Ind. | **74.29**$_{\pm 2.53}$ | 73.90$_{\pm 1.69}$ | 73.65$_{\pm 2.64}$ |
> | UniGCN2[1] | Prod. | 72.94$_{\pm 0.97}$ | 73.29$_{\pm 1.29}$ | **73.58**$_{\pm 1.69}$ |
> |  | Tran. | 72.04$_{\pm 0.84}$ | 72.87$_{\pm 1.12}$ | **73.05**$_{\pm 1.60}$ |
> |  | Ind. | 73.81$_{\pm 2.48}$ | 74.95$_{\pm 2.30}$ | **75.71**$_{\pm 2.54}$ |
> | ED-HNN[2] | Prod. | 72.01$_{\pm 1.06}$ | 73.45$_{\pm 1.48}$ | **73.53**$_{\pm 1.61}$ |
> |  | Tran. | 70.64$_{\pm 1.52}$ | 72.11$_{\pm 1.66}$ | **73.34**$_{\pm 1.78}$ |
> |  | Ind. | 73.79$_{\pm 1.72}$ | **74.79**$_{\pm 1.61}$ | 74.32$_{\pm 1.86}$ |
> | AllSet[3] | Prod. | 71.37$_{\pm 1.60}$ | 73.20$_{\pm 1.96}$ | **73.53**$_{\pm 1.48}$ |
> |  | Tran. | 71.15$_{\pm 1.93}$ | 73.00$_{\pm 2.57}$ | **73.14**$_{\pm 1.32}$ |
> |  | Ind. | 72.97$_{\pm 2.61}$ | **75.36**$_{\pm 2.07}$ | 75.10$_{\pm 2.61}$ |
>
> In the table, UniGCN2 [1], ED-HNN [2], and AllSet [3] are the added "advanced ones" for comparison. We will add those results in Table S9 "Experimental results of LightHGNNs with different teacher architectures." in the Appendix.
>
> [1] Huang et al. Unignn: a unified framework for graph and hypergraph neural networks. IJCAI, 2021.
>
> [2] Wang et al. Equivariant Hypergraph Diffusion Neural Operators, ICLR, 2023.
>
> [3] Chien et al. You are AllSet: A Multiset Function Framework for Hypergraph Neural Networks, ICLR, 2022.

---

> ### Author Response · Authors · 2023-11-19
> **Response to Reviewer tcvF (Part 3/4)**
>
> ### **Response to  W3 and Q3**
> Thanks for your comments. As for the question of *"generalizability of the proposed method across a wider range of scenarios"*, we know that the distillation framework is highly dependent on the task. Different tasks need different distillation strategies. For example, the distillation of image classification[1] is to pull the KL Divergence of the teacher's and student's prediction of the classification, which is not suitable for the image retrieval task[2]. We follow the typical distillation frameworks on graphs [3][4], focusing on the vertex classification task. We thank the reviewer's suggestions. It is crucial to seek a general distillation framework for a wide range of different tasks, like vertex classification, hyperedge classification, and hyperedge prediction. We will consider it in our future work.
>
>
>
> As for the question of *"more types of hypergraph datasets"*, we will add two larger scale hypergraph datasets in the final version, which model the **high-order correlation among foods**. The two datasets [5] both contain eight categories. Specifically, the Recipe-100k dataset contains 101,585 vertices and 12,387 hyperedges, and the Recipe-200k dataset contains 240,094 vertices and 18,129 hyperedges. The vertex features are the bag of words from the sentence that makes the recipe. Hyperedges are the ingredients of the recipe or the Keywords for food preparation steps. Experiments on the two datasets are shown in the following table.
> |  |  | Recipe-100k |  |  | Recipe-200k |  |  |
> |---|---|:---:|:---:|:---:|:---:|:---:|:---:|
> |  |  | #Vertices of Testing | Acc.(%) | Infer Time(s) | #Vertices of Testing | Acc.(%) | Infer Time(s) |
> | HGNN | Trans. | 10,063 | 41.26$_{\pm 3.08}$ | 5.39 | 23,914 |  37.25$_{\pm 4.88}$ | 29.78 |
> |  | Ind. | 90,562 | OOM | $\infty$ | 215,220 | OOM | $\infty$ |
> |  | Prod. | 100,625 | OOM | $\infty$ | 239,134 | OOM | $\infty$ |
> | LightHGNN | Trans. | 10,063 | 41.82$_{\pm 3.48}$ | 0.44 | 23,914 | 38.48$_{\pm 5.59}$ | 0.67 |
> |  | Ind. | 90,562 | 41.15$_{\pm 3.23}$ | 2.18 | 215,220 | 37.62$_{\pm 5.89}$ | 5.05 |
> |  | Prod. | 100,625 | 41.22$_{\pm 3.26}$ | 2.4 | 239,134 | 37.70$_{\pm 5.86}$ | 5.62 |
> | LightHGNN+ | Trans. | 10,063 | **42.50**$_{\pm 3.74}$ | 0.44 | 23,914 | **38.76**$_{\pm 5.24}$ | 0.67 |
> |  | Ind. | 90,562 | **42.28**$_{\pm 3.62}$ | 2.18 | 215,220 | **38.26**$_{\pm 5.67}$ | 5.05 |
> |  | Prod. | 100,625 | **42.31**$_{\pm 3.63}$ | 2.4 | 239,134 | **38.31**$_{\pm 5.62}$ | 5.62 |
>
> As shown in the table, our LightHGNNs can not only achieve better performance but also extremely reduce the inference time. As the scale of the dataset increases, the advantage of our LightHGNNs becomes more obvious. Naive HGNN[6] becomes slower and slower, even throwing Out-Of-Memory errors confronting 10w+ vertices, while our LightHGNNs are still fast. Besides, our LightHGNNs exhibit better performance compared to the naive HGNN under the transductive testing set (about 1w or 2w vertices for testing), which demonstrates the effectiveness of our distillation framework. The results indicate that our LightHGNNs are more suitable for large-scale hypergraph inference.
>
> [1] Hinton et al. Distilling the knowledge in a neural network. NIPS, 2014.
>
> [2] Chen et al. Simplified TinyBERT: Knowledge Distillation for Document Retrieval. ECIR, 2021
>
> [3] Zhang et al. Graph-less neural networks: Teaching old MLPs new tricks via distillation. ICLR, 2022.
>
> [4] Wu et al. Quantifying the knowledge in gnns for reliable distillation into MLPs. ICML, 2023.
>
> [5] Li et al. SHARE: a System for Hierarchical Assistive Recipe Editing. EMNLP, 2022.
>
> [6] Feng et al. Hypergraph Neural Networks. AAAI, 2019.
>
>
> ### **Response to Q4**
> Thanks for your comments. We apologize for the unclear description of how to apply the GNN-distillation methods to hypergraph-structured datasets in the experiments. Since GNN-distillation methods cannot be directly applied to the hypergraph, we first transform the hypergraph into a graph by the typical clique expansion, following [1]. Then, we apply the GNN-distillation methods to the generated graph for training and testing.  We have provided the details in Appendix F.
>
> [1] Gao et al. HGNN$^+$: General Hypergraph Neural Networks. TPAMI, 2023.

---

> ### Author Response · Authors · 2023-11-19
> **Response to Reviewer tcvF (Part 4/4)**
>
> ### **Response to Q5**
> Thanks for your comments. Compared with the distilling graph neural networks (GNN), the challenges of distilling hypergraph neural networks (HGNN) are as follows.
>
> - **The beyond pairwise modeling capability brings more challenges.** The hypergraph is a generalization of the graph, which can model both the low-order structure (graph) and high-order structure. Thus, compared to the distilling graph neural networks that only consider the pair-wise structure, distilling hypergraph neural networks will consider both low-order and high-order structures, which brings more challenges to the distillation of hypergraph neural networks.
> - **The multi-model property of hypergraph brings more challenges.** Hypergraphs can naturally model the multi-modal correlation via the incidence matrix concatenation [1], which is more flexible than the graph. In real-world applications, the complex high-order correlations are inherently multi-modal. For example, in the social network, the people and relationships are represented by vertex and hyperedge, and the people can be divided into different groups, and the relationships can be divided into different types, like family, friends, and colleagues. Clearly, distilling those complex (multi-modal) high-order correlations (hypergraph neural networks) is more challenging than the pairwise correlation (graph neural networks).
> - **The information of the hypergraph is hard to quantify.** As for distilling graph neural networks, the information of the graph is easy to quantify. For example, the degree of the edge is fixed to 2, and the connection of the vertex is simple (single modality and directly propagating messages from other vertices). In contrast, the information in the hypergraph is hard to quantify. For example, the degree of the hyperedge is not fixed (one, two, or more are all legal), and the connection of the vertex is complex (multi-modal and propagating messages from other vertices or hyperedge or group vertices). Thus, controlling the process of distilling hypergraph neural networks or making a reliable hypergraph neural network distillation is more challenging than distilling graph neural networks.
>
> As demonstrated in [2], GCN [3] is a special case of  HGNN confronting the pair-wise correlation. HGNN is more powerful than GNN but also suffers from the higher runtime and memory cost. Thus, to better capture those high-order correlations in the hypergraph, we propose a reliable hyperedge quantification and sampling strategy to achieve topology-aware distillation for hypergraph neural networks. This work is the first attempt at distilling hypergraph neural networks. In the future, we will consider more downstream tasks, more modalities, and more strategies of reliable quantification for distilling hypergraph neural networks.
>
> [1] Feng et al. Hypergraph Neural Networks. AAAI, 2019.
>
> [2] Gao et al. HGNN$^+$: General Hypergraph Neural Networks. TPAMI, 2023.
>
> [3] Kipf et al. Semi-supervised classification with graph convolutional networks. ICLR, 2017.
>
>
> ### **Response to Q6**
> Thanks for your comments. Here, we discuss our significance compared to (Wu et al. 2023)[1]. We will further clarify the difference between our work and KRD[1] and add proper citations in the revised version. The main difference between our work and KRD[1] is two-fold.
> - KRD [1] is a typical reliable-based distillation for graph neural networks. However, its student MLPs can only be supervised by the soft label from those reliable vertices (knowledge points in their paper), which still lose the structure information and cannot be utilized in the hypergraph. However, our LightHGNN$^+$ can quantify the reliability of the high-order correlations and inject the reliable high-order information into the student MLPs via explicit supervision from high-order soft labels of those reliable hyperedges. The proposed LightHGNN$^+$ can sufficiently take advantage of both the vertex features and high-order structure information, thus yielding better performance.
> - Besides, we design a metric to measure the over-smoothing phenomenon (the relevance of features and hypergraph typology) in hypergraph neural networks and empirically demonstrate that our LightHGNN$^+$ has the capability to combat the over-smoothing and achieve better performance, which is not considered in KRD[1].
>
> All those discussions will be added in the revised version.
>
> [1] Wu et al. Quantifying the knowledge in gnns for reliable distillation into MLPs. ICML, 2023.

---

> > ### Author Response · Authors · 2023-11-21
> > **Response to Reviewer tcvF**
> >
> > Dear Reviewer tcvF,
> >
> > We express our gratitude for your valuable time in reviewing our submission. We welcome any further questions you may have regarding our work or rebuttal. Please do not hesitate to share any additional concerns. We are always open to further discussion.
> >
> > Sincerely,
> >
> > Authors.

---

> > > ### Comment · Reviewer_tcvF · 2023-11-21
> > > **Thank you**
> > >
> > > Thank you for the clarifications and efforts for addressing my concerns. I have raised my score.

---

> > > > ### Author Response · Authors · 2023-11-22
> > > > **Thank you**
> > > >
> > > > Thanks for your feedback. We are glad to hear that our explanations have addressed your concerns. We will add those revisions to the camera-ready version.
> > > >
> > > > Thanks again for your careful review and valuable comments to help us improve our submission.

---

### Author Response · Authors · 2023-11-23
**Summary of the Discussion**

Dear Chairs and Reviewers,

Hope this message finds you well.

With the closing of the discussion period, we present a brief summary of our discussion with the reviewers as an overview for reference.
First of all, we thank all the reviewers for their insightful comments and suggestions. We are encouraged that the reviews found our paper is

- R1: pioneer in distilling hypergraph neural networks, well-structured and easy to follow
- R2: interesting, clear, and easy to follow
- R3: inventive problem-solving abilities by filling gaps in existing approaches, well-structured, clearly explained, clearly labeled figures and visualization
- R4: pioneer to explore the knowledge distillation on hypergraphs, adequate experiments, inference efficiency, well-structured and presented

We have carefully read all the comments and responded to them in detail. All of those will be addressed in the final version.

---

We summarize the main concerns of the reviews with the corresponding response as follows.

- **Inference Capability on Large-Scale Hypergraph Datasets.**
We further conduct experiments on two large-scale hypergraph datasets: Recipe-100k (10w vertices) and Recipe-200k (24w vertices). Results show that our LightHGNNs can not only achieve better performance but also extremely reduce the inference time. As the scale of the dataset increases, the advantage of our LightHGNNs becomes more obvious. Naive HGNN becomes slower and slower, even throwing the Out-Of-Memory error confronting 10w+ vertices, while our LightHGNNs are still fast.


- **More SOTA HGNN Methods for Comparison.**
We clarify the proposed LightHGNNs is a general hypergraph neural network distillation framework, which can be applied to any hypergraph neural network to achieve fast and accurate inference. We further conduct experiments on other advanced hypergraph neural networks, including UniGCN2, ED-HNN, and AllSet. Results show as the performance of the teacher HGNN increases, the performance of our LightHGNNs also increases.

- **Novelty of LightHGNNs Compared to Typical Reliable-Based Method KRD.**
We clarify the difference between our work and KRD, as follows.

    - In KRD, its student MLPs can only be supervised by the soft label from those reliable vertices (knowledge point in their paper), which still lose the structure information and cannot be utilized in the hypergraph. However, our LightHGNN+ can quantify the reliability of the high-order correlations and inject reliable high-order information into the student MLPs via explicit supervision from high-order soft labels of those reliable hyperedges. The proposed LightHGNN$^+$ can sufficiently take advantage of both the vertex features and high-order structure information, thus yielding better performance.
    - Besides, we design a metric to measure the over-smoothing phenomenon (the relevance of features and hypergraph typology) in hypergraph neural networks and empirically demonstrate our LightHGNN+ has the capability to combat the over-smoothing and achieve better performance, which is not considered in KRD.


- **Ability of Suppressing Over-smoothing.**
We attempt to increase the number of layers to demonstrate the ability to suppress over-smoothing. Results show that the accuracy of MLP, HGNN, and LightHGNNs all significantly decrease as the number of layers increases, especially for the MLP and HGNN. However, the accuracy of LightHGNN+ is more stable than MLP, HGNN, and LightHGNNs. This is because the proposed LightHGNN+ can alleviate the over-smoothing phenomenon via the hyperedge reliability sampling, thus yielding robust performance combating the increase of the number of layers.

---

Based on the discussion with reviews, we also present a brief summary of our paper as follows.

- **Observation**: Existing hypergraph neural networks suffer from high computational cost, which limits their practical deployment value for larger-scale hypergraph inference.
- **Solution**: We propose a distillation framework to distill hypergraph neural networks into lightweight MLPs, which can achieve fast and accurate inference.
- **Results**: Extensive experiments on 13 real-world graph/hypergraph datasets demonstrate the effectiveness of our distillation framework. Besides, we also conduct experiments on large-scale hypergraph datasets to demonstrate the capability of our LightHGNNs on large-scale hypergraph inference.
- **Highlights**: We quantify the reliability of the high-order correlations and inject the reliable high-order information into the student MLPs via explicit supervision from high-order soft labels of those reliable hyperedges. The proposed LightHGNNs can sufficiently take advantage of both the vertex features and high-order structure information, thus yielding better performance.

Thanks again for your efforts in the reviewing and discussion. We appreciate all the valuable feedback that helped us to improve our submission.

Sincerely

Authors of Paper1556

---

### Meta-Review · Area_Chair_wZi6 · 2023-12-11

**Metareview:**

This paper proposed to distill the hypergraph neural network to an MLP via knowledge distillation to enable efficient inference on the hypergraph using the distilled MLP. To incorporate the topological information into MLP, the authors proposed to construct pseudo labels by aggregating the predictions over reliable hyperedges, which are estimated by the variance of hyperedge predictions under the presence of graph perturbation.

The paper received positive recommendations from all reviewers, who appreciated the problem setting and idea. The primary concerns raised by the reviewers were about (1) clarifications on contributions over KRD, (2) experiments with more teacher HGNNs, and (3) results on larger hypergraph datasets. The authors adequately addressed these concerns in a rebuttal. After reading the paper, review, and rebuttal, AC agrees with the reviewers’ decision and recommends acceptance of the paper. Authors should include the additional experiment results and clarifications added in the rebuttal to the camera-ready version of the paper.

**Justification For Why Not Higher Score:**

The paper is a reasonable extension of KRD to hypergraph.

**Justification For Why Not Lower Score:**

N/A

---

### Decision · Program_Chairs · 2024-01-16

Accept (poster)